# PnP-Flow: Plug-and-Play Image Restoration with Flow Matching

**Ségolène Martin**[1,*]   **Anne Gagneux**[2,*]   **Paul Hagemann**[1,*]   **Gabriele Steidl**[1]

[1] Technische Universität Berlin

[2] ENS de Lyon, CNRS, Université Claude Bernard Lyon 1, Inria, LIP, UMR 5668

* Equal contributions

## Abstract

In this paper, we introduce Plug-and-Play (PnP) Flow Matching, an algorithm for solving imaging inverse problems. PnP methods leverage the strength of pre-trained denoisers, often deep neural networks, by integrating them in optimization schemes. While they achieve state-of-the-art performance on various inverse problems in imaging, PnP approaches face inherent limitations on more generative tasks like inpainting. On the other hand, generative models such as Flow Matching pushed the boundary in image sampling yet lack a clear method for efficient use in image restoration. We propose to combine the PnP framework with Flow Matching (FM) by defining a time-dependent denoiser using a pre-trained FM model. Our algorithm alternates between gradient descent steps on the data-fidelity term, reprojections onto the learned FM path, and denoising. Notably, our method is computationally efficient and memory-friendly, as it avoids backpropagation through ODEs and trace computations. We evaluate its performance on denoising, super-resolution, deblurring, and inpainting tasks, demonstrating superior results compared to existing PnP algorithms and Flow Matching based state-of-the-art methods. Code available at `https://github.com/annegnx/PnP-Flow`.

## 1 Introduction

Image restoration aims to recover an unknown image $x \in \mathbb{R}^d$ from a degraded observation $y \in \mathbb{R}^m$

$$y = Hx + \xi,$$

where $H : \mathbb{R}^d \to \mathbb{R}^m$ is a (linear) degradation operator and $\xi$ describes the underlying additive noise model. Since the problem is usually ill-posed and high dimensional, its treatment is challenging. We assume that the image $x$ is sampled from a random variable $X \in \mathbb{R}^d$ with a density $p_X$, and the observation $y$ from a random variable $Y \in \mathbb{R}^m$ with a density $p_Y$. Then the maximum a posteriori estimator searches for the value with the highest probability of the posterior

$$\arg\max_{x \in \mathbb{R}^d} \left\{ \log p_{X|Y=y}(x) \right\} = \arg\max_{x \in \mathbb{R}^d} \left\{ \log p_{Y|X=x}(y) + \log p_X(x) \right\}, \tag{1}$$

where the first term of the right hand side is the fidelity to the data and the second term represents the prior distribution of the image. Since $p_X$ is generally unknown, and in the absence of training data, it is standard to instead consider a regularized optimization problem of the form:

$$\arg\min_{x \in \mathbb{R}^d} \left\{ F(x) + R(x) \right\}, \tag{2}$$

where $F(x) := -\log p_{Y|X=x}(y)$ and $R : \mathbb{R}^d \to \mathbb{R}$ usually enforces some assumptions on the solution and ensures the existence of a (unique) minimizer. For instance, for Gaussian noise $\mathcal{N}(0, \sigma^2 I_d)$, the data-fidelity corresponds to $F(x) = \frac{1}{2\sigma^2} \|Hx - y\|^2$. The regularized minimization problem in (2) can be efficiently handled using proximal splitting methods (Boyd et al., 2011; Combettes & Pesquet, 2011).

Plug-and-Play (PnP) (Venkatakrishnan et al., 2013) methods build upon the insight that the proximal step on the regularization term is effectively a denoising operation. They can be performed with non-learned denoisers such as BM3D (Chan et al., 2016), however recently these were outperformed by

neural network-based approaches (Meinhardt et al., 2017; Zhang et al., 2017; 2021). While PnP methods have demonstrated considerable effectiveness, recent advances in generative models offer a more sophisticated framework for learning priors directly from data, surpassing the limitations of hand-crafted or neural denoisers.

In particular, generative models are used as regularizers in Bora et al. (2017); Asim et al. (2020); Altekrüger et al. (2023); Wei et al. (2022). Most of these references learn an invertible transport map $T$ between a Gaussian latent distribution and the data distribution, using the change of variables formula to obtain the estimated prior $p_X$ as an explicit function of the known latent density and the map $T$. In particular the normalizing flow network with the Real-NVP (Dinh et al., 2017) architecture is by default invertible and has an easy-to-evaluate log determinant of the Jacobian of $T$. However, this architecture yields rather poor results on large images. Therefore, recent work (Ben-Hamu et al., 2024; Zhang et al., 2024b; Pokle et al., 2024; Chung et al., 2023; Song et al., 2023) focuses more on learning *diffusion* models (Song et al., 2021; Ho et al., 2020; Sohl-Dickstein et al., 2015) or *Flow Matching* models (Lipman et al., 2023; Liu et al., 2023), which do not have these architectural constraints and scale better to large images. The main idea is to establish a path between the latent and the target distribution, which can be optimized in a *simulation-free* way.

In this paper, we focus on Flow Matching models, which learn a velocity field $v : [0,1] \times \mathbb{R}^d \longrightarrow \mathbb{R}^d$ going from the latent to the data distribution. Once the velocity field $v$ is learned, sampling from the target distribution can be done by solving an ODE (Chen et al., 2018)

$$\partial_t f(t,x) = v_t(f(t,x)), \quad f(0,x) = x,$$

where $x$ is sampled from the latent distribution.

However, using Flow Matching to regularize inverse problems such as image restoration tasks is non-trivial. Indeed, when directly solving the MAP problem (1) using the change of variables formula, one faces numerical challenges due to the backpropagation through the ODE. We circumvent this problem by integrating the implicit Flow Matching prior into a custom denoiser that we plug into a Forward-Backward algorithm.

**Contributions.** In this paper, we combine PnP methods with Flow Matching and propose a PnP Flow Matching algorithm. Our contributions are as follows:

- Inspired by optimal transport Flow Matching, we design a time-dependent denoiser based on a pre-trained velocity field $v$ learned through Flow Matching.

- This denoiser is integrated into an adapted Forward-Backward Splitting PnP framework, which cycles through a gradient step on the data-fidelity term, an interpolation step to reproject iterates onto flow trajectories, and a denoising step.

- Our algorithm is simple to implement, requires no backpropagation through the network, and is highly efficient in both memory usage and execution time compared to existing Flow Matching-based methods for solving inverse problems.

- We show that, on two datasets and across denoising, deblurring, inpainting, and super-resolution tasks, our method consistently outperforms state-of-the-art Flow Matching-based and PnP methods in terms of both PSNR and SSIM metrics. The code with all benchmark methods is available at: `https://github.com/annegnx/PnP-Flow` and is now included the DeepInv library[1].

## 2 BACKGROUND

We next provide background on both Plug-and-Play algorithms and Flow Matching models.

### 2.1 PLUG AND PLAY

PnP algorithms were introduced as extensions of proximal splitting methods like Forward-Backward Splitting (FBS) or the alternating direction method of multipliers (ADMM). These are first-order

---

[1]https://github.com/deepinv/deepinv

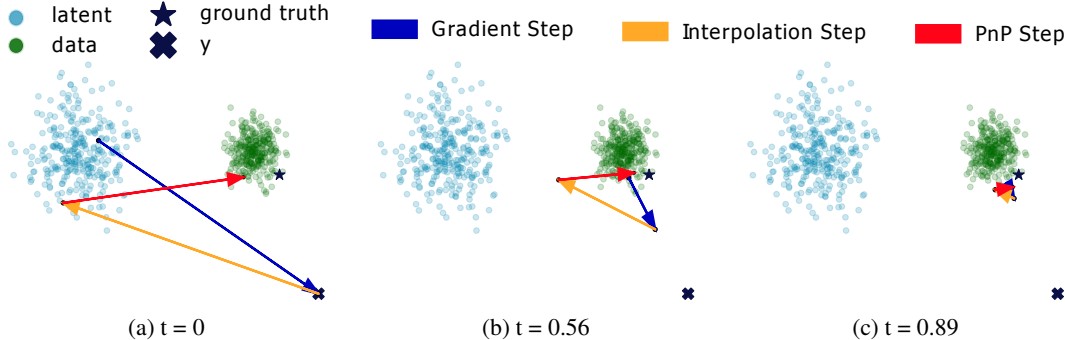

Figure 1: Our method on a 2D denoising task ($\sigma = 1.5$) with Gaussian distributions. An OT Flow Matching model is trained to sample from $P_1 = \mathcal{N}(m, s^2\mathrm{Id})$, with $m = 7$ and $s = 0.5$. At each time step, it performs a standard gradient step on the datafit, followed by a projection onto flow trajectories at time $t$, and finally applies the time-dependent denoiser $D_t$.

optimization algorithms for solving problem (2), alternating between a proximal step on the regularization and a gradient and/or proximal step on the data-fidelity function. The proximal operator $\mathrm{prox}_{\gamma R} : \mathbb{R}^d \to \mathbb{R}^d$, $\gamma > 0$ is defined by $\mathrm{prox}_{\gamma R}(y) := \arg\min_{x \in \mathbb{R}^d} \left\{ \frac{1}{2}\|y - x\|^2 + \gamma R(x) \right\}$. In this paper, we assume that the data-fidelity $F$ is differentiable with Lipschitz continuous gradient and we focus on the FBS algorithm, corresponding to Algorithm 1. The convergence of FBS to a minimizer of $F + R$ is guaranteed as long as the stepsize is chosen as $\gamma \in (0, 1/\mathrm{Lip}(\nabla F))$, where $\mathrm{Lip}(\nabla F)$ is the Lipschitz constant of $\nabla F$ (Combettes & Wajs, 2005).

---

**Algorithm 1: FBS**

**Initialization:** $x^{(0)} \in \mathbb{R}^d, \gamma > 0$
**for** $r = 0, 1, \ldots$ **do**
$\quad\quad z^{(r+1)} = x^{(r)} - \gamma\nabla F(x^{(r)})$
$\quad\quad x^{(r+1)} = \mathrm{prox}_{\gamma R}(z^{(r+1)})$

---

**Algorithm 2: PnP-FBS**

**Initialization:** $x^{(0)} \in \mathbb{R}^d, \gamma > 0$
**for** $r = 0, 1, \ldots$ **do**
$\quad\quad z^{(r+1)} = x^{(r)} - \gamma\nabla F(x^{(r)})$
$\quad\quad x^{(r+1)} = D(z^{(r+1)})$ $\quad\quad\quad$ ▷ PnP step

---

Observing that computing $\mathrm{prox}_{\gamma R}$ corresponds to solving a Gaussian denoising problem with regularization $\gamma R$, Venkatakrishnan et al. (2013) proposed to replace it with an *off-the-shelf* denoiser $D : \mathbb{R}^d \longrightarrow \mathbb{R}^d$, which can be independently designed or learned (see Algorithm 2).

Many PnP-FBS methods, such as Meinhardt et al. (2017); Zhang et al. (2017; 2021); Sun et al. (2019); Terris et al. (2020); Hertrich et al. (2021); Hurault et al. (2022a); Tan et al. (2024) use neural network denoisers. Most of these PnP-FBS algorithms do not converge (Zhang et al., 2017; Sommerhoff et al., 2019) and convergence guarantees usually assume non-expansiveness of the denoiser (Pesquet et al., 2021; Ryu et al., 2019; Hurault et al., 2022b). Yet, constraining the Lipschitz constant of a neural network, e.g., through weight clipping, spectral normalization or averaged operator constructions harms in general its expressiveness (Ryu et al., 2019).

## 2.2 Flow Matching

Let $\mathcal{P}_2(\mathbb{R}^d)$ denote the subspace of probability measures on $\mathbb{R}^d$ having a finite second moment. Let $P_0$ denote the latent probability measure and $P_1$ the target probability measure (i.e., the data distribution). We denote $\cdot_\sharp$ the push-forward operation. We denote $\Gamma(P_0, P_1)$ the set of couplings $\pi \in \mathcal{P}(\mathbb{R}^d \times \mathbb{R}^d)$ having marginals $P_i \in \mathcal{P}_2(\mathbb{R}^d)$, $i = 0, 1$, and $\Gamma_o(P_0, P_1)$ the set of optimal couplings minimizing the Wasserstein-2 distance (Villani, 2008). We now fix a coupling $\pi \in \Gamma(P_0, P_1)$.

The core idea behind Flow Matching is to define a specific target probability path $t \mapsto P_t$, $t \in [0, 1]$, between $P_0$ and $P_1$. Let $P_t := (e_t)_\sharp\pi$, where the map $e_t(x_0, x_1) := (1 - t)x_0 + tx_1$ interpolates between (a latent sample) $x_0$ and (a data sample) $x_1$. The path $P_t$ can be shown to be an absolutely continuous curve, so there exists a Borel vector field $v : [0, 1] \times \mathbb{R}^d \to \mathbb{R}^d$ such that the curve satisfies the *continuity equation*

$$\partial_t P_t + \nabla \cdot (P_t v_t) = 0 \tag{CE}$$

in the sense of distributions (Ambrosio et al., 2008). In addition, there exists a solution $f : [0,1] \times \mathbb{R}^d \to \mathbb{R}^d$ of the ODE

$$\partial_t f(t, x) = v_t(f(t, x)), \quad f(0, x) = x \tag{3}$$

such that $P_t = f(t, \cdot)_\sharp P_0$. If $f$ is known, then a sample $x_1$ from the target distribution $P_1$ can be drawn by first sampling $x_0 \sim P_0$ and then defining $x_1 = f(1, x_0)$.

The goal of Flow Matching is to learn the velocity field of the flow ODE (3). This learning process consists in minimizing the loss function

$$\mathcal{L}_{\text{FM}}(\theta) := \mathbb{E}_{t \sim \mathcal{U}[0,1], x \sim P_t} \left[ \|v_t^\theta(x) - v_t(x)\|^2 \right],$$

where $v^\theta$ is parametrized by a neural network with weights $\theta$. Unfortunately, in practice, the true velocity field $v_t(x)$ is not available. However, Lipman et al. (2023) showed that minimizing $\mathcal{L}_{\text{FM}}(\theta)$ is equivalent to minimizing the Conditional Flow Matching (CFM) loss: $\mathcal{L}_{\text{CFM}}(\theta) = \mathcal{L}_{\text{FM}}(\theta) + \text{const}$, where

$$\mathcal{L}_{\text{CFM}}(\theta) := \mathbb{E}_{t \sim \mathcal{U}[0,1], (x_0, x_1) \sim \pi} \left[ \|v_t^\theta(e_t(x_0, x_1)) - (x_1 - x_0)\|^2 \right]. \tag{4}$$

Minimizing this loss only requires sampling from the coupling $\pi$. For example, if $\pi = P_0 \times P_1$ is the independent coupling, we only need samples from the latent and target distributions. If instead, $\pi \in \Gamma_o(P_0, P_1)$ is an optimal coupling, then we can use a standard optimal transport solver to (approximately) sample from $\pi$, as proposed in Pooladian et al. (2023); Tong et al. (2024).

**Straight-line flows**   Let $v \colon [0,1] \times \mathbb{R}^d \longrightarrow \mathbb{R}$ and let $f$ be a solution to the flow ODE associated with $v$. Given $(X_0, X_1) \sim \pi$, we call $(f, v)$ a *straight-line Flow Matching pair* connecting $X_0$ and $X_1$ if $X_t = f(t, X_0)$ almost surely, where $X_t$ is defined as $X_t := e_t((X_0, X_1)) = tX_1 + (1 - t)X_0$ (Liu et al., 2023; Pooladian et al., 2023). Note that this directly implies $v_t(f(t, X_0)) = \partial_t f(t, X_0) = \partial_t X_t = X_1 - X_0$ almost surely.

We will focus on such straight-line flows later in the paper. Straight or nearly straight paths are preferred since they represent the shortest distance between two points and can be simulated with fewer steps of an ODE solver. This has been explored in many recent works that want to speed up sampling, such as Kornilov et al. (2024); Lee et al. (2023); Yang et al. (2024). A particular case of straight-line pair is given by OT couplings (Pooladian et al., 2023; Tong et al., 2024). Indeed, for $\pi \in \Gamma_o(P_0, P_1)$ with Monge map $T : \mathbb{R}^d \to \mathbb{R}^d$ such that $X_1 = T(X_0)$, there exists a velocity field $v$ for which the pair $(P_t, v_t)$ is a solution of the continuity equation (CE), and such that for all $x \in \mathbb{R}^d$

$$v_t(f(t, x)) = T(x) - x, \tag{5}$$

where $f$ is the solution of the ODE (3), see Ambrosio et al. (2008); Pooladian et al. (2023); Chemseddine et al. (2024) for the precise conditions. By integrating (5), it is clear that $(f, v)$ is a straight-line Flow Matching for the coupling $(X_0, T(X_0))$, since $X_t = (1 - t)X_0 + tX_1 = (1 - t)X_0 + tT(X_0) = X_0 + \int_0^t v_s(f(s, X_0))ds = f(t, X_0)$.

## 3   PNP MEETS FLOW MATCHING

### 3.1   DENOISING OPERATOR FROM FLOW MATCHING

Let $X_0 \sim P_0$ and $X_1 \sim P_1$ with joint distribution $(X_0, X_1) \sim \pi$. Assume we have access to a pre-trained velocity field $v^\theta$. Then we define, for $t \in [0, 1]$, the following time-dependent denoiser

$$D_t := \text{Id} + (1 - t)v_t^\theta, \tag{6}$$

and we propose to use it within a PnP framework.

To motivate the choice of the denoiser in (6), recall that for a fixed time $t \in [0, 1]$, the minimizer $v_t^*$ of the CFM loss (4) over all possible vector fields reads

$$v_t^*(x) = \mathbb{E}[X_1 - X_0 | X_t = x],$$

where $X_t := e_t(X_0, X_1) = (1-t)X_0 + tX_1$ (Benton et al., 2024; Liu et al., 2023). Assume we are in the ideal case where $v_t^\theta = v_t^*$. Then it follows that, for any $x \in \mathbb{R}^d$ and $t \in [0,1]$,

$$D_t(x) = x + (1-t)v_t^*(x)$$
$$= \mathbb{E}_{(X_0, X_1) \sim \pi}[X_t + (1-t)(X_1 - X_0)|X_t = x]$$
$$= \mathbb{E}_{(X_0, X_1) \sim \pi}[X_1|X_t = x].$$

Hence, the operator $D_t$ can be understood at the best approximation of $X_1$ given the knowledge of $X_t$, which is also used in Pokle et al. (2024) and Zhang et al. (2024b).

Just as standard PnP denoisers minimize the MSE loss between noisy and clean samples, the operator $D_t$ is the minimizer of $L^2$-problem $\min_g \mathbb{E}_{(X_0, X_1) \sim \pi}[||X_1 - g(X_t)||^2]$, projecting any noisy point taken along the path onto the target distribution. In particular, the following proposition holds, demonstrating that the best denoising performance is achieved with straight-line flows.

**Proposition 1.** *Assume* $v := v^\theta$ *is continuous and assume that, given* $v$*, the Flow ODE* (3) *has a unique solution* $f$*. Then the denoising loss* $\mathbb{E}_{(X_0, X_1) \sim \pi}[||D_t(X_t) - X_1||^2]$ *is equal to 0 for all* $t \in [0,1]$*, if and only if the couple* $(f, v)$ *is a straight-line Flow Matching pair between* $X_0$ *and* $X_1$.

The proof can be found in Appendix A.1.

As stated in Section 2.2, a special case of straight-line flows is given by OT Flow Matching for which we recover $D_t(X_t) = X_1$. Indeed, in this case, $D_t$ reduces to $T(X_0) = X_1$, with $T$ the Monge map between $P_0$ and $P_1$. We next discuss whether diffusion models induce straight-line flows.

**Remark 2** (Flow Matching versus diffusion models). Contrary to OT Flow Matching, diffusion models, which are also flow ODE methods, do not generally induce straight paths. Indeed, during diffusion training, the target probability path takes the form $\tilde{X}_t = \alpha_t X_0 + \beta_t X_1$ with $\alpha_t \in (0,1)$ and $\beta_t = \sqrt{1 - \alpha_t^2}$ (Song et al., 2021; Liu et al., 2023), which clearly does not match the desired straight path $X_t = (1-t)X_0 + tX_1$. The non-straightness of the flow generated by diffusion models is illustrated in Liu et al. (2023); Liu (2022, Figure 5). On the other hand, a non-straight path obtained with Flow Matching (for instance, using an independent coupling $\pi$) can be *rectified* to a straighter one following the procedure described in Liu et al. (2023).

## 3.2 PNP FLOW MATCHING ALGORITHM

In the previous section, we built a denoiser $D_t$ (defined in (6)). We now want to plug it in a Forward-Backward Splitting algorithm in order to solve inverse problems. Yet, our algorithm differs from the classical PnP-FBS (Algorithm 2) in two key aspects. First, the iterations of the our algorithm depend on time because of the definition of $D_t$. Second, we introduce an intermediate reprojection step between the gradient step on the data-fidelity term and the denoising step. More precisely, at each time $t \in [0,1]$, given the current iterate $x$, our algorithm does the following updates:

1. **Gradient step:** a gradient step on the data-fidelity term, mapping $x$ to $z = x - \gamma \nabla F(x)$ for a given learning rate $\gamma > 0$.

2. **Interpolation step:** In a standard Forward-Backward scheme, the denoiser operator is applied right after the gradient step. Yet, as discussed earlier, our operator $D_t$ was specifically designed to correctly denoise inputs drawn from the straight path $X_t = (1-t)X_0 + tX_1$. If the output $z$ from the gradient step at time $t$ does not lie in the support of $X_t$, there is a high chance that the denoising will not be effective, hence the need to "reproject" it along the flow paths before applying $D_t$. To achieve this, we perform a linear interpolation on $z$, as illustrated in Figure 2: at time $t$, we define $\tilde{z} = (1-t)\varepsilon + tz$, where $\varepsilon$ is a noise sample drawn from $P_0$. A similar idea can be found in (Cheng et al., 2025). Note that while $\varepsilon$ is sampled from $P_0$, it is not necessarily coupled to $z \sim P_1$ via $\pi$. If it were, $D_t$ would map $\tilde{z}$ directly back to $z$, annihilating the denoising effect.

3. **PnP Denoising step**: the operator $D_t$ is applied to the output $\tilde{z}$ of the interpolation step, regularizing the current image by pushing it towards the distribution of $X_1$.

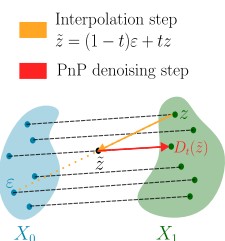

Figure 2: Illustration of the interpolation step.

The resulting discrete-time algorithm is given in Algorithm 3. Figure 1 illustrates the three steps of the algorithm on a denoising problem with a Gaussian prior.

---

**Algorithm 3:** PnP Flow Matching

---

**Input:** Pre-trained network $v^\theta$ by Flow Matching, time sequence $(t_n)_n$ either finite with $t_n = n/N$, $N \in \mathbb{N}$ or infinite with $\lim_{n \to +\infty} t_n = 1$, adaptive stepsizes $(\gamma_n)_n$.
**Initialize:** $x_0 \in \mathbb{R}^d$.
**for** $n = 0, 1, \ldots,$ **do**

$\quad$ $z_n = x_n - \gamma_n \nabla F(x_n)$. $\qquad\qquad\qquad\qquad$ ▷ Gradient step on the data-fidelity term
$\quad$ $\tilde{z}_n = (1 - t_n)\varepsilon + t_n z_n,\ \varepsilon \sim P_0$ $\qquad\qquad\qquad\qquad$ ▷ Linear interpolation
$\quad$ $x_{n+1} = D_{t_n}(\tilde{z}_n)$ $\qquad\qquad\qquad\qquad$ ▷ PnP step with denoiser (6)

**return** $x_{n+1}$

---

**Remark 3** (Averaging in the denoising step). Instead of drawing one noise realization, we can also average over multiple samples $\varepsilon \sim P_0$ in the last step of the algorithm: $x_{n+1} := \mathbb{E}_{\varepsilon \sim P_0}[D_{t_n}(\tilde{z}_n(\varepsilon))]$ with $\tilde{z}_n(\varepsilon) := (1 - t_n)\varepsilon + t_n z_n$. The algorithm's output is then deterministic. In practice, averaging over a few realizations slightly improves the numerical results.

**Time dependent learning rate** Using a constant learning rate independent of time can give too much importance to the data fit. For example, if $\gamma_t = 1$ for all $t \in [0, 1]$ on a simple denoising task, the algorithm will return the noisy sample $y$ since $D_1 = \mathrm{Id}$. To prevent this, $\gamma_t$ should decrease with $t$ to balance the contributions of the datafit and the denoiser. We set $\gamma_t = (1 - t)^\alpha$ with $\alpha \in (0, 1]$ for the remainder of the paper. This choice yields good numerical results in our experiments, but alternative values for $\gamma_t$ could also be explored.

**Convergence** Assuming that the sequence produced by the algorithm is bounded in the infinite time regime, we have the following convergence result. The proof can be found in Appendix A.2.

**Proposition 4.** *Assume that $F : \mathbb{R}^d \to \mathbb{R}$ is continuously differentiable and that the learned vector field $v : [0, 1] \times \mathbb{R}^d \to \mathbb{R}^d$ is continuous. Assume $P_0$ has bounded support. Let the time sequence $(t_n)_{n \in \mathbb{N}}$ satisfy $\sum_{n=0}^{\infty}(1 - t_n) < +\infty$ and let $\gamma_n := 1 - t_n$, $n \in \mathbb{N}$. If the sequence $(x_n)_{n \in \mathbb{N}}$ obtained by Algorithm 3 is bounded, then it converges.*

## 4 RELATED WORK

Our approach combining PnP with Flow Matching relates to several existing methods.

**Pre-trained Flow Matching methods** Using pre-trained Flow Matching models for regularizing image inverse problems has been the focus of several recent works.

OT-ODE (Pokle et al., 2024) assumes a Gaussian latent distribution and uses Tweedie's formula to derive a new velocity field $\tilde{v}_t(x, y) = \mathbb{E}[X_1 - X_0 \mid X_t = x, Y = y]$ from the original velocity field $v_t(x) = \mathbb{E}[X_1 - X_0 \mid X_t = x]$, all without requiring retraining. In practice, they sample from the posterior distribution $X_1 \mid Y$ by solving the ODE with the new velocity field using an Euler scheme.

In Zhang et al. (2024b), the authors introduce Flow-Priors, a method to tackle the MAP problem by approximating it as a sequence of time-dependent MAP subproblems. Using Tweedie's formula, they show that for $t < 1$, the gradient of the distribution $P_t$ of $X_t$ can be computed in closed form, allowing for the use of gradient descent to optimize these subproblems. However, the closed-form expression for the gradient relies on the assumption of an independent coupling $\pi$ and a Gaussian latent distribution. Besides, their method requires computing $\mathrm{Tr}\, \nabla v^\theta$, which is expensive.

In Ben-Hamu et al. (2024), an implicit regularization approach called D-Flow is considered: instead of minimizing the Gaussian data-fidelity function $x \mapsto \|Hx - y\|^2$, they minimize the latent loss $z \mapsto \|H(f(1, z)) - y\|^2$, where $f$ is a solution to the flow ODE given the pre-trained network. The two problems are theoretically equivalent since $f(1, \cdot)$ is invertible. However, since the latent loss is not convex, first- or second-order optimization methods may not find the global minimizer. Interestingly, this is beneficial because the true minimizer of the original problem is simply the

pseudo-inverse, which may not be desirable. The authors optimize the latent loss by backpropagating through the ODE solution with a few Euler steps, though this remains computationally expensive.

**(PnP) Diffusion methods**    While we present the first PnP method based on Flow Matching, related works combine diffusion models with the PnP framework. We refer to Appendix A.12 for further discussion.

In Zhu et al. (2023), a Half Quadratic Splitting algorithm is used, alternating between a proximal step on the data-fidelity term and a proximal step on the regularization. Following the PnP strategy, the proximal step for the regularization term is replaced with a denoising step using a pre-trained diffusion model. The denoiser they use is reminiscent of the one we use, where the velocity is replaced by the gradient of the score function. Their method also includes an interpolation step with random noise, mapping the estimated image at each iteration back to the diffusion path. In Garber & Tirer (2024), the proximal step on the data-fidelity term in Zhu et al. (2023) is traded for a gradient step. Finally, conditional image restoration is explored in (Graikos et al., 2022), which uses a more relaxed definition of "plug and play". They integrate a pre-trained diffusion model under different conditions, leading to a variational objective similar to methods like Mardani et al. (2024).

## 5 NUMERICAL EXPERIMENTS

### 5.1 BASELINES

We benchmark our method against three state-of-the-art Flow Matching-based restoration methods: OT-ODE (Pokle et al., 2024), D-Flow (Ben-Hamu et al., 2024) and Flow Priors (Zhang et al., 2024b). As no official implementations were publicly available for these methods, we developed our own based on the descriptions provided in their respective publications. We have made every effort to ensure faithful implementations, included in the code attached to this paper. We also benchmark our method against PnP-Diff (Zhu et al., 2023), a PnP algorithm based on diffusion models. Additionally, we compare our approach with the state-of-the-art PnP-FBS (Hurault et al., 2022a).

### 5.2 EXPERIMENTAL SETUP

**Datasets**    We evaluate all methods on two datasets: CelebA (Yang et al., 2015), with images resized to $128 \times 128$, and AFHQ-Cat, a subset of the Animal FacesHQ dataset (Choi et al., 2020) focused on the *cat* class, with images resized to $256 \times 256$. All images are normalized to the range $[-1, 1]$. For CelebA, we use the standard training, validation, and test splits. For AFHQ-Cat, as no validation split is provided, we randomly select 32 images from the test set to create a validation set.

**Models**    For each dataset, we trained a Flow Matching model from scratch using the Mini Batch OT Flow Matching approach (Tong et al., 2024), as this choice of coupling usually leads to straight paths. We used a standard Gaussian as the latent distribution and a U-Net (Ronneberger et al., 2015) taken from Huang et al. (2021); Ho et al. (2020) as the model. The training parameters were a learning rate of $10^{-4}$, 200 epochs with a batch size of 128 for CelebA, and 400 epochs with a batch size of 64 for AFHQ-Cat. We train the denoising network for the PnP method PnP-GS (Hurault et al., 2022a) employing the same U-Net architecture. For PnP-Diff, because training a diffusion model with the same U-Net architecture as Flow-Matching yielded poor results due to insufficient number parameters, we used the pre-trained model from Choi et al. (2021), implemented in the DeepInv library (Tachella et al., 2023). Note that the pre-trained diffusion model was trained on the FFHQ dataset (Karras et al., 2019), making the comparison indirect, but we had no alternative.

**Settings for the experiments**    We evaluate the methods using 100 test images across five restoration problems: denoising with Gaussian noise ($\sigma = 0.2$); deblurring using a $61 \times 61$ Gaussian kernel ($\sigma_b = 1.0$ for CelebA, $\sigma_b = 3.0$ for AFHQ-Cat); super-resolution ($2\times$ downsampling for CelebA, $4\times$ for AFHQ-Cat); box-inpainting with a centered $s \times s$ mask ($s = 40$ for CelebA, $s = 80$ for AFHQ-Cat); and random pixel inpainting (70% masked pixels). For deblurring, super-resolution, and box-inpainting, we add Gaussian noise with $\sigma = 0.05$, and for random inpainting $\sigma = 0.01$.

**Hyper-parameters** We optimize the hyper-parameters for each method using a grid search on the validation set, selecting the configuration that yields the highest Peak Signal-to-Noise Ratio (PSNR). The optimal values identified for each dataset and problem scenario are detailed in Appendix A.8. Our proposed method has two hyper-parameters: the exponent $\alpha$ in the learning rate schedule $\gamma_n = (1-t_n)^\alpha$, and the number of uniformly spaced time steps was set to $N = 100$ for most experiments. We averaged the results of the denoising step over 5 realizations of the interpolation step.

## 5.3 MAIN RESULTS

We report benchmark results for all methods across several restoration tasks, measuring average PSNR and Structural Similarity (SSIM) on 100 test images. To ensure reproducibility, all experiments were seeded. Results are presented in Table 1 for CelebA and Table 2 for AFHQ-Cat. "N/A" indicates cases where the method is inapplicable; for instance, PnP-GS (Hurault et al., 2022a) is not designed for generative tasks like box inpainting, and PnP-Diff relies on a diffusion model trained on another dataset, making its evaluation on box inpainting inappropriate.

The tables show that our method consistently ranks first or second in both reconstruction metrics across all tasks and datasets. More importantly, it demonstrates stability across tasks, unlike other methods. For example, D-Flow and Flow-Priors perform well in box inpainting but struggle with denoising, while PnP-GS, PnP-Diff, and OT-ODE excel in denoising and deblurring but perform worse on pseudo-generative tasks.

In terms of visual quality (Fig. 3, Fig. 4, and Appendix A.7), our method produces realistic, artifact-free images, though sometimes slightly over-smoothed. While D-Flow generates realistic images, it occasionally suffers from hallucinations (e.g., eye color shifts in CelebA denoising tasks). Flow-Priors introduces noise and artifacts, while OT-ODE captures textures well but struggles with image generation. Finally, Appendix A.6 shows the progression of the reconstruction with respect to time.

Table 1: Comparisons of state-of-the-art methods on different inverse problems on the dataset CelebA. Results are averaged across 100 test images.

| Method | Denoising $\sigma = 0.2$ | | Deblurring $\sigma = 0.05, \sigma_b = 1.0$ | | Super-res. $\sigma = 0.05, \times 2$ | | Rand. inpaint. $\sigma = 0.01, 70\%$ | | Box inpaint. $\sigma = 0.05, 40 \times 40$ | |
|---|---|---|---|---|---|---|---|---|---|---|
| | PSNR | SSIM | PSNR | SSIM | PSNR | SSIM | PSNR | SSIM | PSNR | SSIM |
| Degraded | 20.00 | 0.348 | 27.67 | 0.740 | 10.17 | 0.182 | 11.82 | 0.197 | 22.12 | 0.742 |
| PnP-Diff | 31.00 | 0.883 | 32.49 | 0.911 | 31.20 | 0.893 | 31.43 | 0.917 | N/A | N/A |
| PnP-GS | **32.45** | 0.908 | 33.65 | 0.924 | 30.69 | 0.889 | 28.45 | 0.848 | N/A | N/A |
| OT-ODE | 30.50 | 0.867 | 32.63 | 0.915 | 31.05 | 0.902 | 28.36 | 0.865 | 28.84 | 0.914 |
| D-Flow | 26.42 | 0.651 | 31.07 | 0.877 | 30.75 | 0.866 | 33.07 | 0.938 | 29.70 | 0.893 |
| Flow-Priors | 29.26 | 0.766 | 31.40 | 0.856 | 28.35 | 0.717 | 32.33 | 0.945 | 29.40 | 0.858 |
| PnP-Flow (ours) | **32.45** | **0.911** | **34.51** | **0.940** | **31.49** | **0.907** | **33.54** | **0.953** | **30.59** | **0.943** |

Table 2: Comparisons of state-of-the-art methods on different inverse problems on the dataset AFHQ-Cat. Results are averaged across 100 test images.

| Method | Denoising $\sigma = 0.2$ | | Deblurring $\sigma = 0.05, \sigma_b = 3.0$ | | Super-res. $\sigma = 0.05, \times 4$ | | Rand. inpaint. $\sigma = 0.01, 70\%$ | | Box inpaint. $\sigma = 0.05, 80 \times 80$ | |
|---|---|---|---|---|---|---|---|---|---|---|
| | PSNR | SSIM | PSNR | SSIM | PSNR | SSIM | PSNR | SSIM | PSNR | SSIM |
| Degraded | 20.00 | 0.319 | 23.77 | 0.514 | 11.59 | 0.216 | 13.35 | 0.234 | 21.50 | 0.744 |
| PnP-Diff | 30.27 | 0.835 | **27.97** | **0.764** | 23.22 | 0.601 | 31.08 | 0.882 | N/A | N/A |
| PnP-GS | **32.34** | **0.895** | 27.33 | 0.749 | 21.86 | 0.619 | 29.61 | 0.855 | N/A | N/A |
| OT-ODE | 29.90 | 0.831 | 26.43 | 0.709 | 25.17 | 0.711 | 28.84 | 0.838 | 23.88 | 0.874 |
| D-Flow | 26.22 | 0.620 | 27.49 | 0.740 | 24.10 | 0.595 | 31.37 | 0.888 | 26.69 | 0.833 |
| Flow-Priors | 29.32 | 0.768 | 25.78 | 0.692 | 23.34 | 0.573 | 31.76 | 0.909 | 25.85 | 0.822 |
| PnP-Flow (ours) | 31.65 | 0.876 | 27.62 | 0.763 | **26.75** | **0.774** | **32.98** | **0.930** | **26.87** | **0.904** |

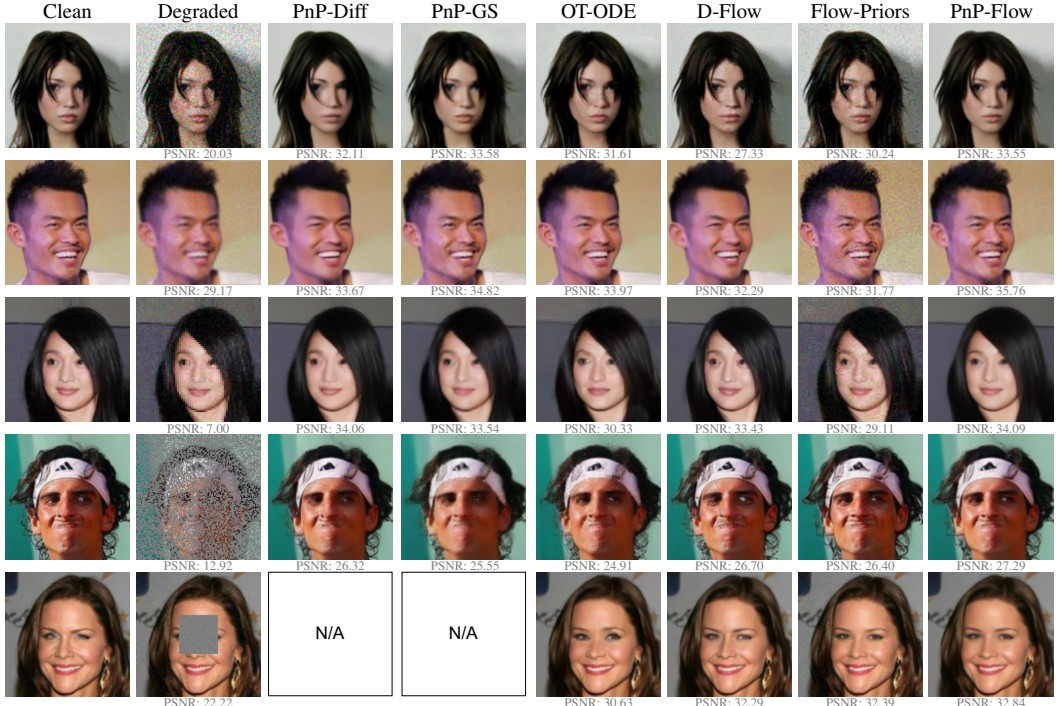

Figure 3: Comparison of image restoration methods on CelebA: denoising (1st row), Gaussian deblurring (2nd row), super-resolution (3rd row), random pixel inpainting (4th row), box-inpainting (5th row). N/A means "method not applicable". Zoom in to see that PnP-Flow performs consistently well across all tasks compared to the baselines.

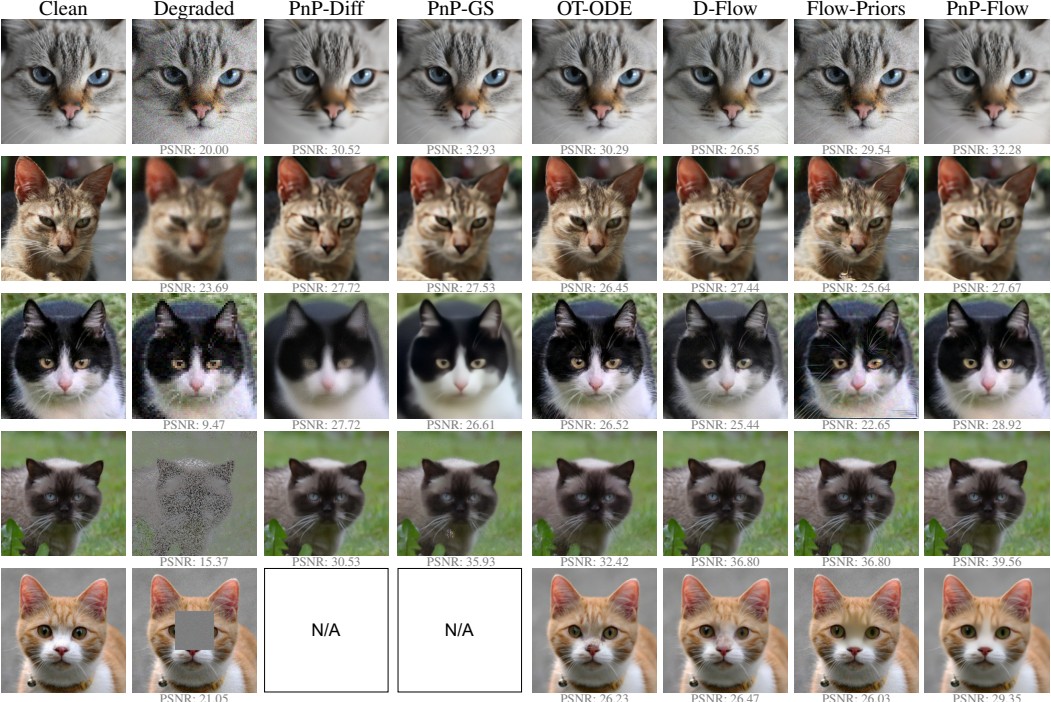

Figure 4: Comparison of image restoration methods on AFHQ-Cat: denoising (1st row), Gaussian deblurring (2nd row), super-resolution (3rd row), random pixel inpainting (4th row), box-inpainting (5th row). N/A means "method not applicable".

## 5.4 PRACTICAL ASPECTS

**Computation time and memory** All experiments in this section are conducted on a single NVIDIA RTX 6000 Ada Generation with 48GB RAM. We measure the averaged time per image to complete a deblurring task on the CelebA dataset. We also compute the peak GPU memory load per image. The results are averaged over 100 images (25 batches of 4 images each). We use the same settings as those used for reporting performance metrics on CelebA. Results are in Table 3.

Table 3: Time and memory metrics per image on the deblurring task on CelebA ($128 \times 128$).

| Method | Computation time | GPU peak memory load |
|---|---|---|
| OT-ODE | 1.50s | 0.65GB |
| Flow-Priors | 16.01s | 2.96GB |
| D-Flow | 32.19s | 5.91GB |
| PnP-Flow (ours) | 3.40s | 0.10GB |

**Sensitivity to initialization** Notably, our algorithm does not rely on a good initialization. By starting the algorithm at time $t_0 = 0$, the linear interpolation step initially outputs a pure noise, which is then given to the denoiser. As a result, the algorithm's performance is independent of the initialization. In Appendix A.4, we stress that this is not the case for other methods.

**Choice of the latent distribution** Our algorithm can be used with any latent distribution. This is in the spirit of Flow Matching models (Lipman et al., 2023), which do not rely on a Gaussian latent as opposed to diffusion models. In particular, recently there has been a trend of modelling categorical data with Flow Matching (Boll et al., 2024; Stark et al., 2024; Davis et al., 2024). Our method does not rely on a Gaussian latent, contrary to the other Flow Matching restoration methods. As an example to illustrate the performance of our approach in a non Gaussian case, in Appendix A.3 we conduct an experiment with a Dirichlet latent distribution, inspired by (Stark et al., 2024).

**Adaptability to any straight-line flows** Our denoiser is rooted in the straight-line flow framework, which motivates our use of OT Flow Matching. However, other choices of FM models are possible. Notably, Rectified Flows (Liu et al., 2023) are of particular interest, as the method described allows for the straightening of any flow model. In Appendix A.5, we show how our method performs similarly to what we observed in Section 5.3 using pre-trained Rectified Flows.

## 6 CONCLUSION

We introduced PnP-Flow Matching, and compared it to recent Flow Matching and PnP methods. A great strength of our method is its versatility: it requires few hyperparameters, uses minimal memory and computational resources, delivers very good performance across various inverse problems, and supports different latent distributions as well as flexible initialization. Regarding limitations, our reconstructions seem to be more on the smooth side, which relates to the fact that our denoiser is mean square estimator. This however is a common tradeoff in regularization of inverse problems (Blau & Michaeli, 2018). Next, it would be valuable to apply our method to other types of measurement noise, such as Poisson noise (Hurault et al., 2023). Preliminary results with Laplace noise are provided in Appendix A.11. Furthermore, our method could be explored in the context of blind inverse problems, as suggested by the promising results in Appendix A.14. Another potential improvement would be to leverage different latent distributions (Boll et al., 2024; Gat et al., 2024; Stark et al., 2024) for modeling categorical distributions. In particular, Dirichlet distributions are widely used in biological and molecular data analysis, making this a relevant domain for further investigation. Finally, an ambitious question for future research would be to explore how the PnP-flow algorithm could be adapted into a posterior sampling algorithm.

**Acknowledgements**  Ségolène Martin's work is carried out in the framework of the DFG funded Cluster of Excellence EXC 2046 MATH+ (project ID: AA5-8). Anne Gagneux thanks the RT IASIS for supporting her work through the PROSSIMO grant. Anne Gagneux also acknowledges support from the Berlin Mathematical School of MATH+, which funded her research visit to TU Berlin. Paul Hagemann acknowledges funding by the German Research Foundation (DFG) within the project SPP 2298 "Theoretical Foundations of Deep Learning". The authors thank the Blaise Pascal Center for the computational means. It uses the SIDUS (Quemener & Corvellec, 2013) solution.

The authors would like to thank Yasi Zhang for helping with the Flow priors code, Bastian Boll for helpful discussions and Marien Renaud for spotting some mistakes.

**Ethics Statement**  Our method makes use of pre-trained generative models, such that the general concerns of such models apply. In particular, these generative models carry inherent biases and can potentially be misused.

However, our work is foundational and we believe there are many relevant applications, such as medical imaging or scientific applications which outweigh the negative usages. Furthermore our work involves only small Flow Matching models, which carry very low risk of being used for malicious purposes.

**Reproducibility Statement**  We implemented all the baselines and release the code in the supplementary material. In Appendix A.8 we state the hyper-parameter search procedure and the found values. Further, all our theoretical results state the precise assumptions and contain full proofs.

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

# A APPENDIX

## A.1 PROOF OF PROPOSITION 1

*Proof.* We have by the definition of the denoiser (6) (it was also observed in Lee et al. (2023))

$$\mathbb{E}_{(X_0,X_1)\sim\pi}[\|D_t(X_t) - X_1\|^2] = (1-t)^2\mathbb{E}_{(X_0,X_1)\sim\pi}[\|v_t(X_t) - (X_1 - X_0)\|^2].$$

If we assume that the denoising loss is zero, this yields for all $t \in [0,1)$ that $v_t(X_t) = X_1 - X_0$ almost surely. By continuity this also follows for $t = 1$. By the same arguments as in (Liu et al., 2023, Theorem 3.6 iii)+ iv)) we have that both $t \mapsto f(t, X_0)$ and $t \mapsto X_t$ are solution to the flow ODE initialized at $X_0$, since

$$X_t = X_0 + tv_t(X_t) = X_0 + \int_0^t v_s(X_s)ds \quad \text{a.s.}$$

where we used that $v$ is constant with time. This implies that $\partial_t X_t = v_t(X_t)$ almost surely. Together with the uniqueness of the ODE solution, it follows that $f(t, X_0) = X_t$ almost surely.

On the other hand, if $(f, v)$ is a straight-line Flow Matching pair connecting $X_0$ and $X_1$, then we obtain that

$$v_t(X_t) = v_t(f(t, X_0)) = \partial_t f(t, X_0) = \partial_t X_t = X_1 - X_0 \quad \text{a.s.}$$

Therefore the denoising loss is zero. $\qquad\square$

## A.2 PROOF OF PROPOSITION 4

We provide the proof for Proposition 4 here below. Note that the assumption of noise boundedness can be easily satisfied by clipping the noise. Additionally, it is naturally ensured numerically since floating-point numbers have finite precision.

*Proof.* By the algorithm and definition of $D_{t_n}$, we have

$$x_{n+1} = D_{t_n}(u_n) = u_n + (1 - t_n)v_{t_n}^\theta(u_n),$$

where together with the definition of $\gamma_n$,

$$u_n := (1 - t_n)\varepsilon_n + t_n(x_n - \gamma_n\nabla F(x_n)) = t_n x_n + (1 - t_n)(\varepsilon_n - t_n\nabla F(x_n)).$$

Hence we obtain

$$\|x_{n+1} - x_n\| = (1 - t_n)\|\varepsilon_n - x_n - t_n\nabla F(x_n) + v_{t_n}^\theta(u_n)\|.$$

By assumption on $F$ and $v$ and since both $(x_n)_n$ and the noise $\varepsilon_n$ are bounded, the expression in the norm is bounded as well, say by $M > 0$. Then, by assumption on $t_n$, we conclude

$$\sum_{n=0}^\infty \|x_{n+1} - x_n\| = M\sum_{n=0}^\infty (1 - t_n) < \infty,$$

so that $(x_n)_n$ is a Cauchy sequence and converges. $\qquad\square$

## A.3 NUMERICAL RESULTS USING A LATENT DIRICHLET DISTRIBUTION

One of the main practical advantages of Flow Matching over diffusion is that one can choose a latent distribution with is not Gaussian. Motivated by DNA design and other discrete data, there has been growing interest in using a Dirichlet latent distribution in Flow Matching (Stark et al., 2024; Boll et al., 2024; Davis et al., 2024). Here, we want to show that applying PnP-Flow with a Flow Matching model trained on a Dirichlet latent distribution still yields good reconstructions. For this, we use the MNIST dataset (LeCun et al., 1998), rescaling each image to lie on the simplex, and train a Flow Matching model. We then apply this model to inpainting and super-resolution tasks and present the results. Importantly, the goal of this experiment is not to achieve state-of-the-art performance, but to illustrate that our algorithm generalizes to different latent distributions.

For this, we train a Dirichlet Flow Matching model for 200 epochs with a standard OT Flow Matching loss and a Dirichlet distribution with parameters $(1, ..., 1) \in \mathbb{R}^{784}$. This latent distribution is also used in Stark et al. (2024) and amounts to the uniform distribution on the 784-dimensional simplex. Note that the interpolation step now involves Dirichlet noise instead of Gaussian, as this noise is drawn according to $P_0$. We reconstruct our images with $\gamma = 1$ and 300 steps. Remarkably, almost no modifications for the algorithm are needed. In particular, the generated images $x$ almost lie perfectly on the simplex without any normalization. We show the images in Figure 5. We compare to (Ben-Hamu et al., 2024), where we adapt the regularization in the algorithm so that the optimized variable $z$ lies on the simplex, i.e., we use the loss

$$L(z) = \|y - H(f(1, z))\|^2 + \lambda\|(\sum_{i,j} z_{i,j}) - 1)\|^2,$$

for a single latent image $z$, where $f(1, z)$ is realized doing 5 Euler steps using the mid point rule and $\lambda$ is a regularization constant.

As one can see, the PnP Flow outperforms D-Flow on all tasks. Note that the other Flow Matching baselines (Pokle et al., 2024; Zhang et al., 2024b) are not usable since their algorithm heavily depends on Gaussian paths.

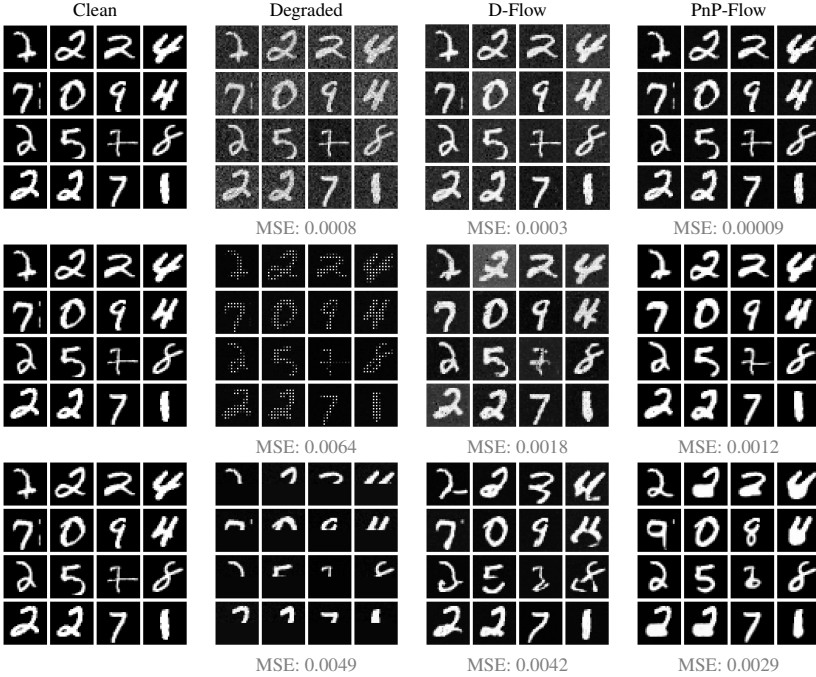

Figure 5: Dirichlet Flow Matching experiment on Simplex-MNIST, for denoising (1st row), super-resolution (2nd row), box-inpainting (3rd row). We measure the reconstruction error as the mean L2 distance (called MSE) between ground truth and reconstruction averaged over the 16 images.

### A.4    Sensitivity to initialization

Interestingly, our method is inherently independent of the algorithm's initialization. For the image restoration problems considered in this work, initialization is not so much of a concern because a reasonable starting point for the solution $x$ is given by $H^\top y$, where $y$ is the observation and $H$ is the degradation operator. However, for more complex problems where $H^\top y$ is far from resembling a natural image, such as in CT reconstruction (Guo et al., 2016) or phase retrieval (Manekar et al., 2020), this property becomes much more relevant.

In contrast, competing methods are more sensitive to initialization and cannot be started from any value. As recommended in the paper, our implementation of OT-ODE is initialized with $t_0 y + (1 - t_0)\epsilon$ where $\epsilon \sim \mathcal{N}(0, I_d)$ and $t_0$ is the initialization time. The latent variable in D-Flow is initialized

as $\alpha T^{-1}(H^\top y) + (1-\alpha)\epsilon$, where $T^{-1}$ is the reverse flow, $\epsilon \sim \mathcal{N}(0, I_d)$ is a random Gaussian noise, and $\alpha \in (0,1)$ is a blending coefficient. Flow-Priors, on the other hand, is initialized with random noise and, like our method, does not depend on a "good" initialization.

In Figure 6, we illustrate the impact of changing the standard initialization for all methods to a black image on a Gaussian deblurring task, comparing the robustness of each approach to poor initialization.

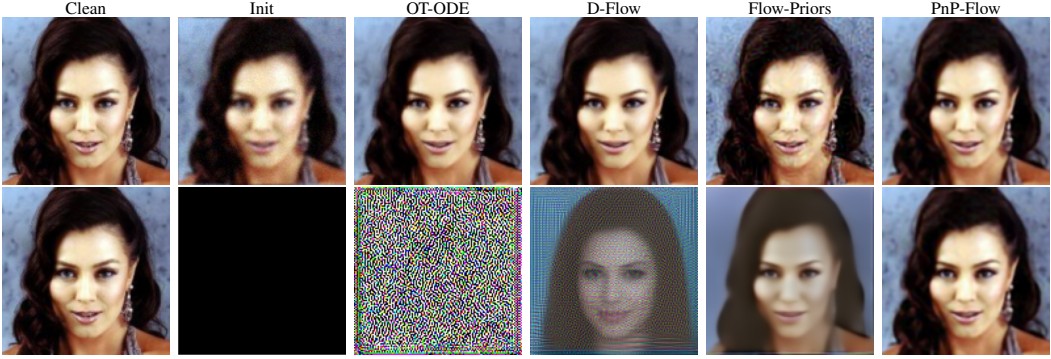

Figure 6: Comparison of restoration methods on the CelebA dataset, for a Gaussian deblurring task and for two different initializations: default initialization recommended for each method (1st row), initialization set to the zero image (2rd row).

## A.5 EXPERIMENTS WITH RECTIFIED FLOWS

The image restoration experiments we carry out in the core of the paper use a pre-trained OT Flow Matching model. Yet, the theory behind our method holds for any straight-line Flow Matching model. In this section, we show how our method PnP-Flow compares to Flow Priors and OT-ODE using, for all three methods, a pre-trained Rectified Flow model Liu et al. (2023) on CelebA-HQ dataset (Karras et al., 2018) with image dimension $256 \times 256$. We use the checkpoint provided by the Rectified Flow repository[2]. We were not able to run D-Flow using the Rectified Flow model. Quantitative results are in Table 4 and qualitative results on paintbrush inpainting are displayed in Figure 7.

Table 4: Comparisons of state-of-the-art methods on different inverse problems on the dataset CelebA-HQ. Results are averaged across 100 test images.

| Method | Denoising $\sigma = 0.2$ | | Deblurring $\sigma = 0.05, \sigma_b = 3.0$ | | Super-res. $\sigma = 0.05, \times 4$ | | Rand. inpaint. $\sigma = 0.01, 70\%$ | | Box inpaint. $\sigma = 0.05, 80 \times 80$ | |
|---|---|---|---|---|---|---|---|---|---|---|
| | PSNR | SSIM | PSNR | SSIM | PSNR | SSIM | PSNR | SSIM | PSNR | SSIM |
| Degraded | 20.00 | 0.277 | 23.94 | 0.514 | 9.98 | 0.040 | 12.28 | 0.197 | 22.14 | 0.726 |
| OT-ODE | 29.38 | 0.731 | 25.47 | 0.589 | 24.24 | 0.578 | 26.59 | 0.639 | 26.68 | 0.683 |
| Flow-Priors | 25.73 | 0.524 | 28.09 | 0.804 | 23.64 | 0.494 | 33.26 | **0.935** | 29.89 | 0.817 |
| PnP-Flow (ours) | **32.65** | **0.891** | **28.57** | **0.815** | **25.92** | **0.762** | **33.60** | **0.935** | **31.11** | **0.928** |

## A.6 PROGRESSION OF THE PNP-FLOW RECONSTRUCTION WITH TIME

Figure 8 presents the progression of the reconstruction outputed by the PnP-Flow with respect to time.

---

[2]https://github.com/gnobitab/RectifiedFlow

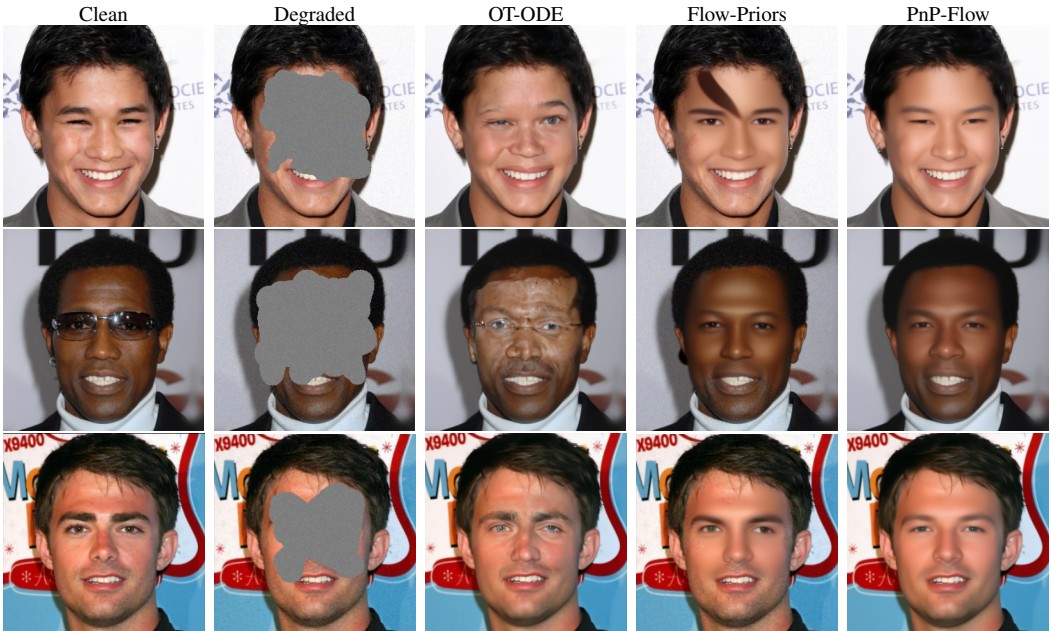

Figure 7: Comparison of restoration methods on the CelebA-HQ dataset for paintbrush inpainting using Rectified Flow model.

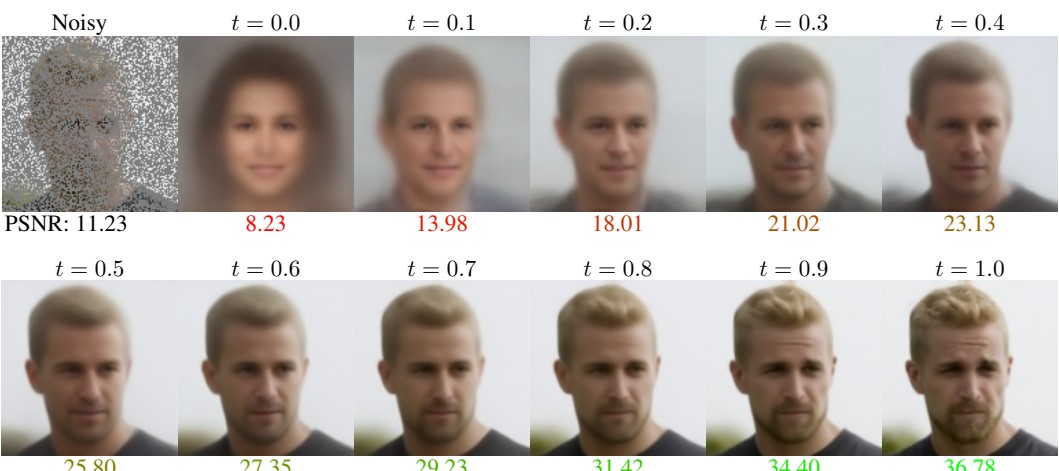

Figure 8: Results for random inpainting using PnP-Flow across different iterations (time steps) with corresponding PSNR values. As expected, in the early iterations, the output resembles a natural face but diverges from the noisy observation. PSNR improves progressively with each iteration.

## A.7 ADDITIONAL VISUAL RESULTS

Here, we provide additional visual results, where we take the same ground truth image for the different inverse problems, see Fig. 9 and Fig. 10 for CelebA dataset and Fig. 11 and Fig. 12 for AFHQ-cat dataset.

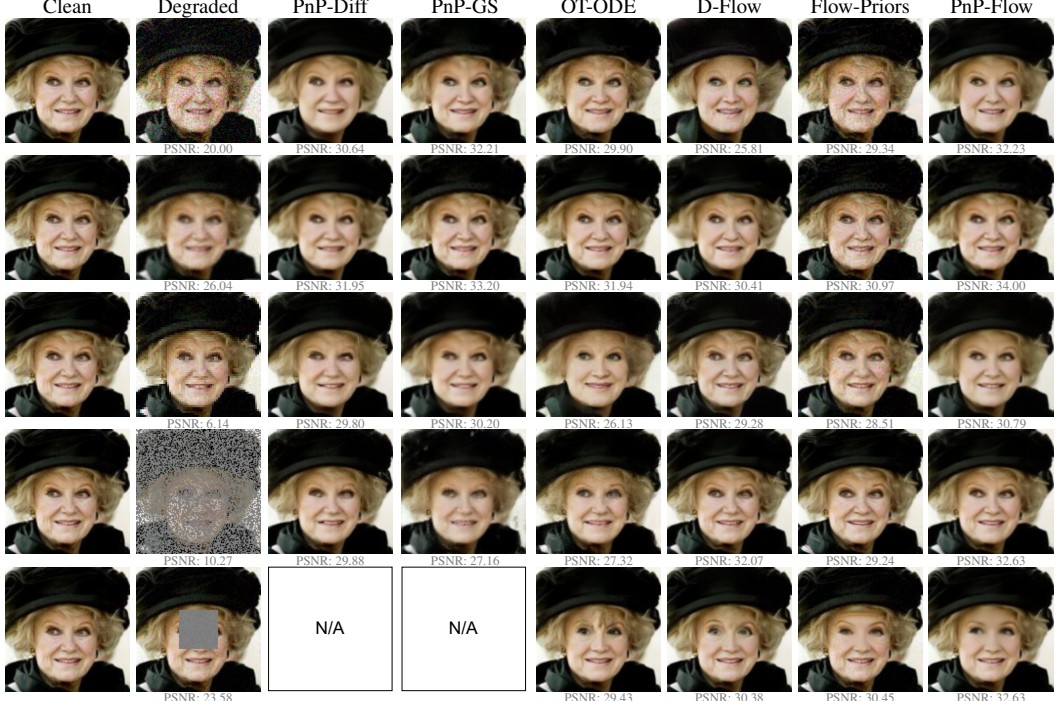

Figure 9: Comparison of image restoration methods on CelebA: denoising (1st row), Gaussian deblurring (2nd row), super-resolution (3rd row), free-form inpainting (4th row).

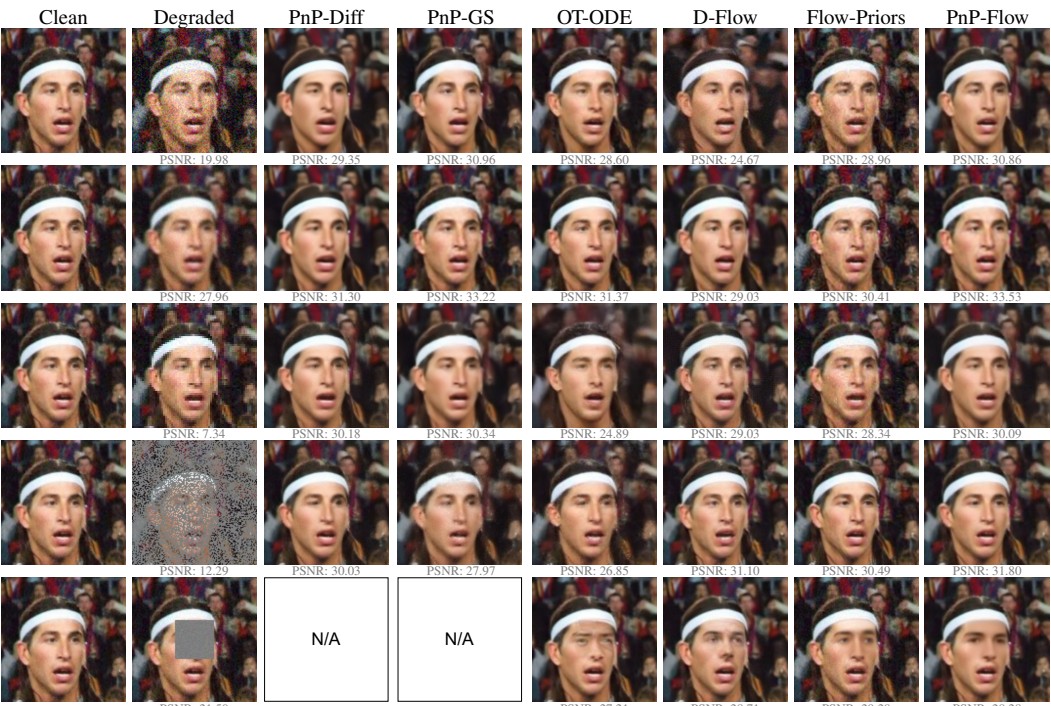

Figure 10: Comparison of image restoration methods on CelebA: denoising (1st row), Gaussian deblurring (2nd row), super-resolution (3rd row), free-form inpainting (4th row).

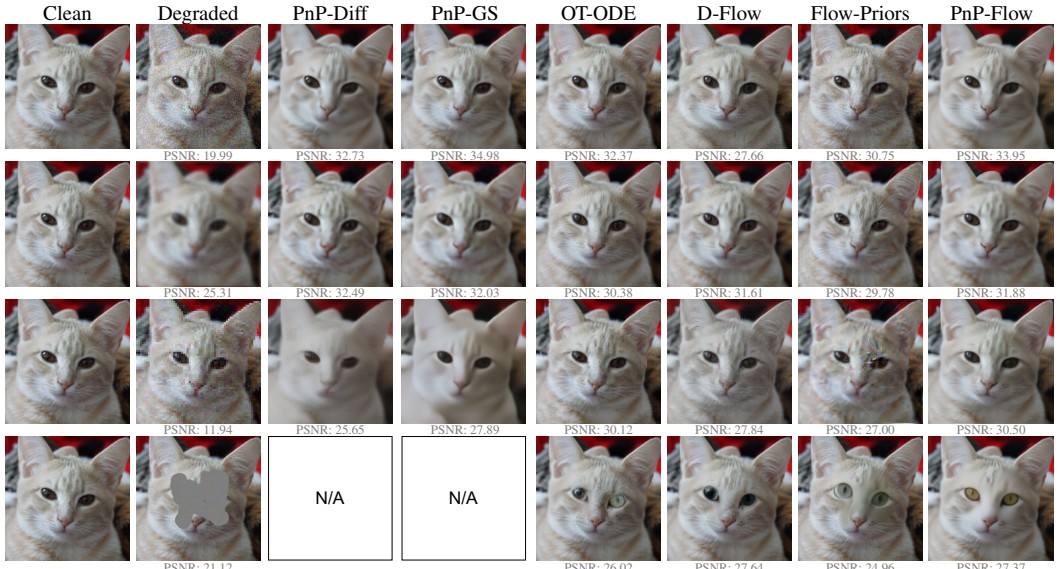

Figure 11: Comparison of restoration methods on AFHQ-Cat: denoising (1st row), Gaussian deblurring (2nd row), super-resolution (3rd row), free-form inpainting (4th row).

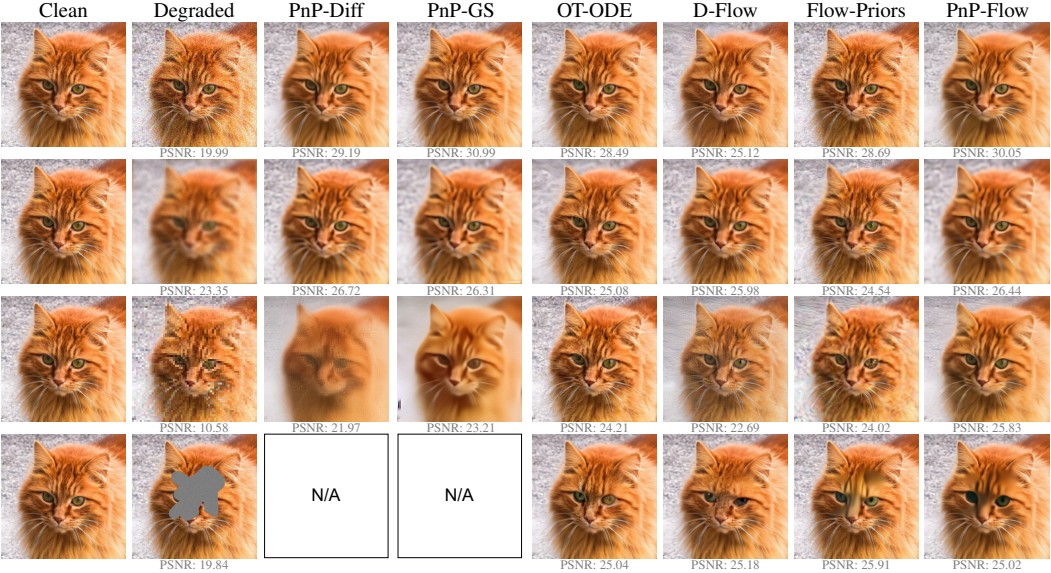

Figure 12: Comparison of image restoration methods on AFHQ-Cat: denoising (1st row), Gaussian deblurring (2nd row), super-resolution (3rd row), free-form inpainting (4th row).

## A.8 Optimal hyper-parameters values

We implemented the Gradient Step denoiser from Hurault et al. (2022a;b) following the repository[3]. For random inpainting, the PnP algorithm is the Half Quadratic Splitting with the parameters given in the paper (Hurault et al., 2022a). For denoising, we only apply the trained denoiser once, feeding it with the true noise level $\sigma$. For all other methods, we use Proximal Gradient Descent (PGD), as prescribed in Hurault et al. (2022b). Note that we do not constrain the Lipschitz constant of the denoiser. For the PGD case, we tune three hyper-parameters: the learning rate $\gamma \in \{0.99, 2.0\}$ in the gradient step (which is related to the regularization parameter $\lambda$ in the paper), the inertia parameter $\alpha \in \{0.3, 0.5, 0.8, 1.0\}$ which coresponds to the relaxed denoiser $D_\sigma^\alpha = \alpha D_\sigma + (1-\alpha)\text{Id}$, and the factor $\sigma_f \in \{1., 1.2, 1.5, 1.8, 2., 3., 4., 5.\}$ so that the denoiser gets as noise map $\sigma_f \times \sigma$ (where $\sigma$ is the true noise level). We also considered the number of iterations as a hyperparameter that we tuned on the validation set (with maximum number of iterations fixed to 100).

For PnP-Diff, we tuned two parameters: the regularization parameter $\lambda \in \{1.0, 5.0, 10.0, 100.0, 1000.0\}$ and the blending parameter $\zeta \in \{0.1, 0.3, 0.5, 1.0\}$. The number of iterations was fixed to 100.

For D-Flow, we adjusted the blending parameter for initialization $\alpha \in \{0.1, 0.3, 0.5\}$ and the regularization parameter $\lambda \in \{0.1, 0.01, 0.001\}$. We observed that the PSNR of the reconstruction did not consistently increase across iterations (the method does not always converge); thus, we fine-tuned the value of the last iteration while keeping it below 20 for computational efficiency. The number of iterations for the inner LBFGS optimization was set to 20, and the number of Euler step when solving the ODE to 5, as recommended in the paper Ben-Hamu et al. (2024).

For OT-ODE, we tuned the initial time $t_0 \in \{0.1, 0.2, 0.3, 0.4\}$ and the type of learning step $\gamma$ (either $\sqrt{t}$ or constant). The number of iterations was set to 100.

For Flow-Priors, we considered -as described in the paper- the two hyper-parameters $\eta \in \{10^{-3}, 10^{-2}, 10^{-1}\}$ and $\lambda \in \{10^2, 10^3, 10^4, 10^4\}$ which respectively correspond to the step size for the gradient descent and the guidance weight (weight put on the data likelihood). The number of iterations was set to 100 and the number of iner iterations to $K = 1$.

Finally, for our method PnP-Flow, we adjusted the exponent in the learning rate $\alpha \in \{0.01, 0.1, 0.3, 0.5, 0.8, 1.0\}$ and the number of time steps $N \in \{100, 200, 500\}$. When increasing $N$ beyond 100 resulted in less than a 0.2 dB improvement in PSNR, we set $N = 100$ for computational efficiency. We average the output of the denoising step in Algorithm 3 over 5 realizations of the interpolation step. To speed up the algorithm, this number can be reduced to 1 with only a minor impact on performance.

## A.9 Visualization of each step of the algorithm

In Figure 13, we show the output of the PnP-Flow algorithm at different stages for an image from the CelebA dataset: after the gradient step, the interpolation step, and the denoising step.

## A.10 Additional evaluation metric

Here, we add a comparison of the models in the LPIPS metric Zhang et al. (2018), which measures the feature distance between the ground truth and generated images via some pretrained classification network. The results are contained in Table 7 and Table 8. None of the methods stands out as a clear winner with respect to the LPIPS metric. However, our PnP-Flow shows a good performance, often being the second or third best method. In Blau & Michaeli (2018), it is argued that perceptual and distortion seem to be at odds with one another: indeed, one can show that the optimal distortion performance (optimizing the PNSR) does not yield optimal perceptual performance (more in line with LPIPS).

---

[3]https://github.com/samuro95/GSPnP

Table 5: Hyper-parameters used for all methods on the CelebA dataset. The values were selected based on the highest PSNR on the validation split.

|  | Denoising | Deblurring | Super-res. | Rand. inpaint. | Box inpaint. |
|---|---|---|---|---|---|
| **PnP-Diff** | | | | | |
| $\zeta$ (blending) | 1.0 | 1.0 | 1.0 | 1.0 | N/A |
| $\lambda$ (regularization) | 1.0 | 1000.0 | 100.0 | 1.0 | N/A |
| **PnP-GS** | | | | | |
| $\gamma$ (learning rate) | - | 2.0 | 2.0 | - | N/A |
| $\alpha$ (inertia param.) | 1.0 | 0.3 | 1.0 | - | N/A |
| $\sigma_f$ (factor for noise input) | 1.0 | 1.8 | 3.0 | - | N/A |
| $n_{\text{iter}}$ (number of iter.) | 1 | 35 | 20 | 23 | N/A |
| **OT-ODE** | | | | | |
| $t_0$ (initial time) | 0.3 | 0.4 | 0.1 | 0.1 | 0.1 |
| $\gamma$ | $\sqrt{t}$ | $\sqrt{t}$ | constant | constant | $\sqrt{t}$ |
| **Flow-Priors** | | | | | |
| $\lambda$ (regularization) | 100 | 1,000 | 10,000 | 10,000 | 10,000 |
| $\eta$ (learning rate) | 0.01 | 0.01 | 0.1 | 0.01 | 0.01 |
| **D-Flow** | | | | | |
| $\lambda$ (regularization) | 0.001 | 0.001 | 0.001 | 0.01 | 0.001 |
| $\alpha$ (blending) | 0.1 | 0.1 | 0.1 | 0.1 | 0.1 |
| $n_{\text{iter}}$ (number of iter.) | 3 | 7 | 10 | 20 | 9 |
| **PnP-Flow** | | | | | |
| $\alpha$ (learning rate factor) | 0.8 | 0.01 | 0.3 | 0.01 | 0.5 |
| $N$ (Number of time steps) | 100 | 100 | 100 | 100 | 100 |

Table 6: Hyper-parameters used for all methods on the AFHQ-Cat dataset. The values were selected based on the highest PSNR on the validation split.

|  | Denoising | Deblurring | Super-res. | Rand. inpaint. | Box inpaint. |
|---|---|---|---|---|---|
| **PnP-Diff** | | | | | |
| $\zeta$ (blending) | 1.0 | 1.0 | 1.0 | 1.0 | N/A |
| $\lambda$ (regularization) | 1.0 | 1000.0 | 100.0 | 1.0 | N/A |
| **PnP-GS** | | | | | |
| $\gamma$ (learning rate) | - | 2.0 | 2.0 | - | N/A |
| $\alpha$ (inertia param.) | 1.0 | 0.3 | 1.0 | - | N/A |
| $\sigma_f$ (factor for noise input) | 1.0 | 1.8 | 5.0 | - | N/A |
| $n_{\text{iter}}$ (number of iter.) | 1 | 60 | 50 | 23. | N/A |
| **OT-ODE** | | | | | |
| $t_0$ (initial time) | 0.3 | 0.3 | 0.1 | 0.1 | 0.1 |
| $\gamma$ | $\sqrt{t}$ | $\sqrt{t}$ | constant | constant | $\sqrt{t}$ |
| **Flow-Priors** | | | | | |
| $\lambda$ (regularization) | 100 | 1,000 | 10,000 | 10,000 | 10,000 |
| $\eta$ (learning rate) | 0.01 | 0.01 | 0.1 | 0.01 | 0.01 |
| **D-Flow** | | | | | |
| $\lambda$ (regularization) | 0.001 | 0.01 | 0.001 | 0.001 | 0.01 |
| $\alpha$ (blending) | 0.1 | 0.5 | 0.1 | 0.1 | 0.1 |
| $n_{\text{iter}}$ (number of iter.) | 3 | 20 | 20 | 20 | 9 |
| **PnP-Flow** | | | | | |
| $\alpha$ (learning rate factor) | 0.8 | 0.01 | 0.01 | 0.01 | 0.5 |
| $N$ (Number of time steps) | 100 | 500 | 500 | 200 | 100 |

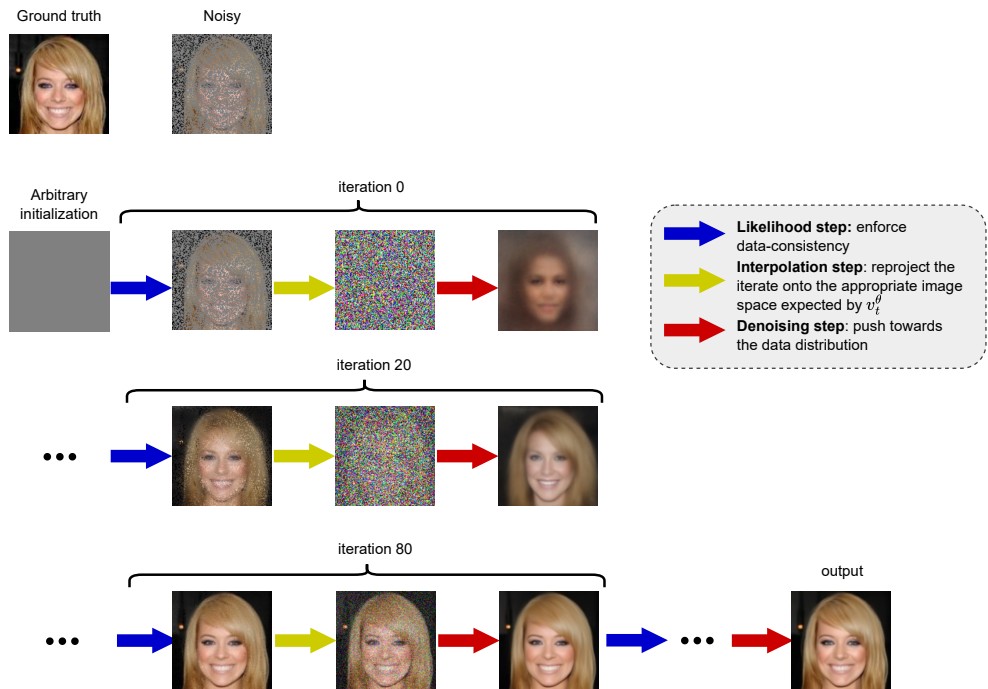

Figure 13: Progression of the output of the algorithm at intermediate steps (log-likelihood step, reprojection onto the path, and denoising) for a random inpainting task.

Table 7: Comparisons of state-of-the-art methods on different inverse problems on the dataset CelebA using the LPIPS perceptual metric Zhang et al. (2018). Results are averaged across 100 test images.

| Method | Denoising $\sigma = 0.2$ | Deblurring $\sigma = 0.05, \sigma_{\mathrm{b}} = 1.0$ | Super-res. $\sigma = 0.05, \times 4$ | Rand. inpaint. $\sigma = 0.01, 70\%$ | Box inpaint. $\sigma = 0.05, 40 \times 40$ |
|---|---|---|---|---|---|
| Degraded | 0.373 | 0.125 | 0.827 | 1.034 | 0.213 |
| PnP-Diff | 0.060 | 0.060 | 0.034 | 0.025 | N/A |
| PnP-GS | 0.035 | 0.040 | 0.053 | 0.085 | N/A |
| OT-ODE | 0.110 | 0.096 | 0.091 | 0.137 | 0.102 |
| D-Flow | 0.078 | 0.053 | 0.027 | 0.021 | 0.035 |
| Flow-Priors | 0.137 | 0.056 | 0.101 | 0.018 | 0.048 |
| PnP-Flow (ours) | 0.056 | 0.046 | 0.055 | 0.021 | 0.043 |

## A.11 HANDLING NON-GAUSSIAN NOISE MODELS

Our approach is capable of handling various types of degradation noise, beyond Gaussian. In general, any noise model corresponding to a differentiable likelihood car be handled by our method effortlessly. Note that this is not the case of all Flow-Matching methods. In particular, the methods D-Flow and OT-ODE are designed under the assumption of standard Gaussian noise. Although it may be possible to adapt these methods to other kinds of noises, doing so would require additional modifications, which we leave for future work.

In the following experiment, we consider a super-resolution task with a degradation model incorporating Laplace noise:

$$y = \mathrm{Laplace}(Hx),$$

Table 8: Comparisons of state-of-the-art methods on different inverse problems on the dataset AFHQ-Cat using the LPIPS perceptual metric Zhang et al. (2018). Results are averaged across 100 test images.

| Method | Denoising $\sigma = 0.2$ | Deblurring $\sigma = 0.05, \sigma_{\rm b} = 3.0$ | Super-res. $\sigma = 0.05, \times 4$ | Rand. inpaint. $\sigma = 0.01, 70\%$ | Box inpaint. $\sigma = 0.05, 80 \times 80$ |
|---|---|---|---|---|---|
| Degraded | 0.524 | 0.460 | 0.888 | 1.069 | 0.206 |
| PnP-Diff | 0.194 | 0.354 | 0.438 | 0.064 | N/A |
| PnP-GS | 0.075 | 0.360 | 0.384 | 0.116 | N/A |
| OT-ODE | 0.190 | 0.226 | 0.193 | 0.226 | 0.187 |
| D-Flow | 0.177 | 0.178 | 0.207 | 0.049 | 0.091 |
| Flow-Priors | 0.159 | 0.191 | 0.279 | 0.047 | 0.122 |
| PnP-Flow (ours) | 0.170 | 0.341 | 0.258 | 0.043 | 0.118 |

where $H : \mathbb{R}^d \to \mathbb{R}^m$ is 2-downsampling linear operator. Laplace noise, which has heavier tails than Gaussian noise, is well-suited for modeling data with outliers. In this context, the data-fidelity function is defined as $f(x) = \|Hx - y\|_1$ and is differentiable almost everywhere.

For each method, we tune the hyper-parameters via grid search, selecting the configuration that yields the highest PSNR, similar to our previous experiments. We report the results in terms of PSNR, SSIM, and LPIPS in Table 9. We also show the visual results in Figure 14.

Table 9: Comparisons of state-of-the-art methods on Laplace super-resolution experiments on the dataset CelebA. Results are averaged across 100 test images.

| Method | Laplace super-resolution $\sigma = 0.1$ | | | Laplace super-resolution $\sigma = 0.3$ | | |
|---|---|---|---|---|---|---|
| | PSNR | SSIM | LPIPS | PSNR | SSIM | LPIPS |
| Degraded | 8.82 | 0.086 | 0.845 | 8.76 | 0.040 | 0.865 |
| PnP-Diff | 29.67 | 0.866 | **0.059** | 24.82 | 0.730 | 0.268 |
| PnP-GS | 28.50 | 0.828 | 0.079 | 25.31 | 0.725 | 0.185 |
| Flow-Priors | 27.53 | 0.721 | 0.102 | 21.72 | 0.454 | 0.376 |
| PnP-Flow (ours) | **30.30** | **0.895** | 0.063 | **26.50** | **0.809** | **0.108** |

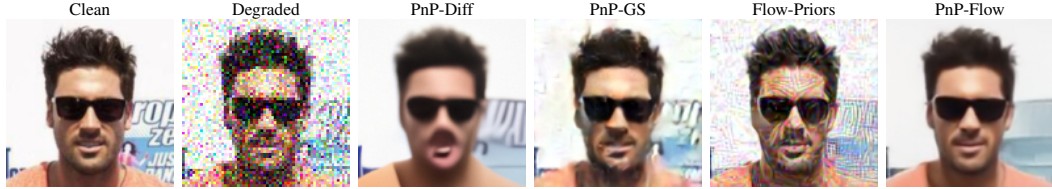

| Clean | Degraded | PnP-Diff | PnP-GS | Flow-Priors | PnP-Flow |

Figure 14: Example of reconstructions obtained on a super-resolution tasks with Laplace noise ($\sigma = 0.3$), on the CelebA dataset.

## A.12 DISCUSSION OF RELATED DIFFUSION APPROACHES

In He et al. (2024), based on the seminal work Chung et al. (2023), a similar algorithm is proposed in the diffusion case (although for latent models): Our algorithm closely resembles their pixel space algorithm with a different noise update rule. However, note that they do not relate it to PnP methods, and that we outlined several theoretical and practical advantages of flow matching over diffusion models. Further similar ideas are explored in Wu et al. (2024); Zhang et al. (2024a); Xu & Chi (2024); Li et al. (2024), which focus on posterior sampling using diffusion models. These works

employ a similar decoupling of the likelihood (data-consistency) sampling step and a denoising diffusion sampler (referred to as the *decoder* in Li et al. (2024)) that acts as an implicit image prior. Additionally, Wang et al. (2023) introduces a particularly efficient null-space approach. Instead of making steps on the log likelihood like Chung et al. (2023), they optimize in the null space of the linear operator directly. Such ideas could in principle also be applied to our algorithm, which we leave for future work.

## A.13 ADDITIONAL EXPERIMENTS WITH DIFFUSION-BASED METHODS

Table 10: Comparison of PnP-Flow with diffusion-based methods on various inverse problems using the CelebA dataset. Results are averaged over 100 test images.

| Method | Denoising $\sigma = 0.2$ | | | Deblurring $\sigma = 0.05$, $\sigma_b = 1.0$ | | | Super-resolution $\sigma = 0.05$, $\times 2$ | | | Random Inpainting $\sigma = 0.01$, 70% mask | | |
|---|---|---|---|---|---|---|---|---|---|---|---|---|
| | PSNR | SSIM | LPIPS | PSNR | SSIM | LPIPS | PSNR | SSIM | LPIPS | PSNR | SSIM | LPIPS |
| Degraded | 20.00 | 0.348 | 0.373 | 27.67 | 0.740 | 0.125 | 7.53 | 0.012 | 0.827 | 11.82 | 0.197 | 1.034 |
| PnP-Diff | 31.00 | 0.883 | 0.060 | 32.49 | 0.911 | 0.060 | 31.20 | 0.893 | **0.034** | 31.43 | 0.917 | 0.025 |
| DDRM | 31.46 | 0.895 | 0.071 | 31.44 | 0.901 | 0.080 | **31.56** | **0.909** | 0.046 | 30.31 | 0.902 | 0.040 |
| DPS | 31.52 | 0.887 | 0.068 | 31.42 | 0.917 | **0.038** | 29.18 | 0.841 | 0.058 | 26.83 | 0.858 | 0.122 |
| PnP-Flow (Ours) | **32.45** | **0.911** | **0.056** | **34.51** | **0.940** | 0.046 | 31.49 | 0.907 | 0.055 | **33.54** | **0.953** | **0.021** |

Table 11: Comparison of PnP-Flow with diffusion-based methods on various inverse problems using the AFHQ-Cat dataset. Results are averaged over 100 test images.

| Method | Denoising $\sigma = 0.2$ | | | Deblurring $\sigma = 0.05$, $\sigma_b = 3.0$ | | | Super-resolution $\sigma = 0.05$, $\times 4$ | | | Random Inpainting $\sigma = 0.01$, 70% mask | | |
|---|---|---|---|---|---|---|---|---|---|---|---|---|
| | PSNR | SSIM | LPIPS | PSNR | SSIM | LPIPS | PSNR | SSIM | LPIPS | PSNR | SSIM | LPIPS |
| Degraded | 20.00 | 0.319 | 0.524 | 23.77 | 0.514 | 0.460 | 10.74 | 0.042 | 0.888 | 13.35 | 0.234 | 1.069 |
| PnP-Diff | 30.27 | 0.835 | 0.194 | **27.97** | **0.764** | 0.354 | 23.22 | 0.601 | 0.438 | 31.08 | 0.882 | 0.064 |
| DDRM | 30.60 | 0.838 | 0.240 | 25.89 | 0.680 | 0.430 | 27.50 | 0.746 | 0.329 | 30.24 | 0.857 | 0.109 |
| DPS | 27.64 | 0.743 | 0.327 | 24.31 | 0.673 | 0.390 | **27.83** | 0.755 | 0.322 | 27.23 | 0.739 | 0.334 |
| PnP-Flow (Ours) | **31.65** | **0.876** | **0.170** | 27.62 | 0.763 | **0.341** | 26.75 | **0.774** | **0.258** | **32.98** | **0.930** | **0.043** |

In Table 10 and Table 11, we provide extensive benchmarks using diffusion-based methods (PnP-Diff Zhu et al. (2023), DDRM Kawar et al. (2022) and DPS Chung et al. (2023) ), respectively on the CelebA and AFHQ-Cat datasets. Qualitative results are given in Figure 15. All diffusion-based methods use the same pre-trained model from Choi et al. (2021), implemented in the DeepInv library[4](Tachella et al., 2023). Note that the pre-trained diffusion model was trained on the FFHQ dataset (Karras et al., 2019), making the comparison indirect. For all methods, a gridsearch over hyperparameters has been conducted (same methodology as followed for the other experiments, see Appendix A.8). More precisely, for DDRM, we adjusted the value of $\eta$ and $\eta_B$ with a grid search over $\{0.7, 0.8, 0.9, 1.0\}$ following the ablation study in Kawar et al. (2022). For DPS, we adjusted the value of $\eta \in \{0.5, 0.7, 0.8, 0.85, 0.9, 0.95\}$ which controls the stochasticity.

## A.14 EXTENSION TO BLIND INVERSE PROBLEMS

In this Section, we describe how our algorithm can be used for blind deconvolution Levin et al. (2009), i.e., in a deblurring problem, where the forward operator is unknown. Here, we assume that we know the kernel size and that the kernel is the same across channels. The true forward operator is given by a Gaussian blur and we attempt to simultaneously reconstruct blur and "true" image. This we do as follows: We loop over a discrete time sequence and after each time-step in Algorithm 3, we run a gradient descent update of the "learned" kernel on the log likelihood loss $\|y - H_{\text{learned}}(x)\|^2$ with the Adam optimizer (Kingma & Ba, 2017). The result can be seen for one examplary image in Figure 16, where we see that our reconstruction is clearly improving upon the blurry measurement. In Figure 17, we repeat this experiment for a random inpainting task, learning

---

[4]https://github.com/deepinv/deepinv

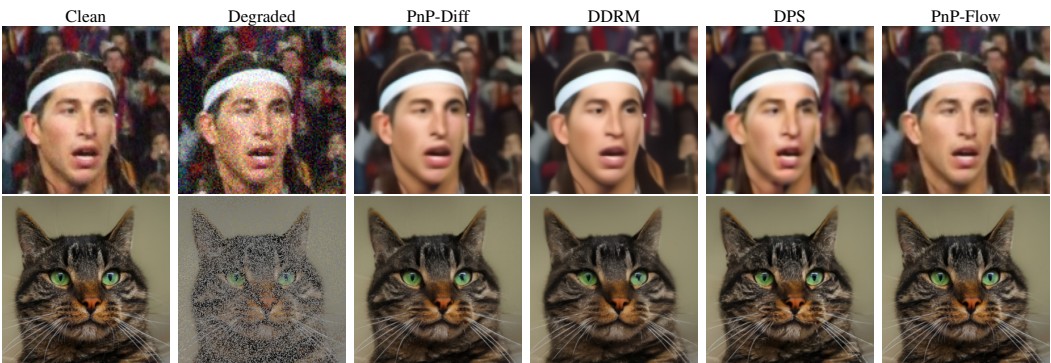

Figure 15: Up: example of reconstructions obtained on a Gaussian denoising task ($\sigma = 0.2$) on the CelebA dataset. Bottom: example of reconstructions obtained on a random inpainting task (70% of masked pixels) on the AFHQ-Cat dataset.

both the reconstruction and the parameterized mask. This seems to be an interesting avenue for future research.

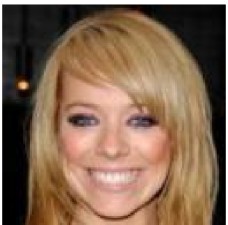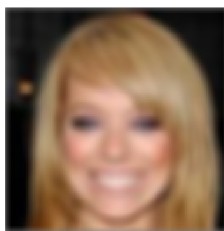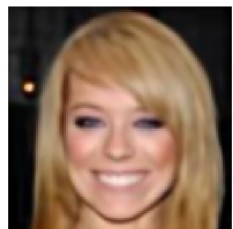
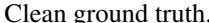

Clean ground truth.          Blurry measurement.          Reconstruction with our method.

Figure 16: Application of our method to a blind deconvolution problem.

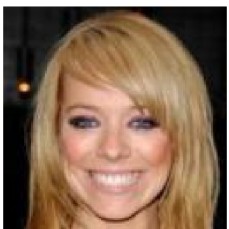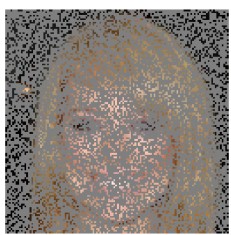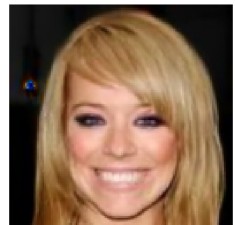

Clean ground truth.          Blurry measurement.          Reconstruction with our method.

Figure 17: Application of our method to a blind masking problem.

