# OpenReview forum: "PnP-Flow: Plug-and-Play Image Restoration with Flow Matching"
_ICLR.cc/2025/Conference — ICLR 2025 Poster_

### Official Review · Reviewer_VJZf · 2024-11-03

**Soundness:** 3
**Presentation:** 3
**Contribution:** 3
**Rating:** 8
**Confidence:** 3

**Summary:**

In this paper, the authors proposed a plug-and-play image restoration method based on flow matching. The reformulation starts from the forward-backward splitting algorithm, where the proximal step is replaced by a denoising step to form the plug-and-play forward-backward splitting algorithm. The authors insert a specific flow matching method, namely straight-line flows into the PnP-FBS framework due to the computation efficiency of the straight-line flows. Formally, the PnP flow matching algorithm consists of three steps: a gradient step on the data fidelity term, an interpolation step, and a PnP denoising step that is specifically designed to denoise inputs drawn from the straight path.

**Strengths:**

1. A new plug-and-play method based on flow matching is proposed in this paper.

2. The paper is well-written.

3. The derivations in this paper are rigorous.

4. The computational complexity and memory footprint of the proposed method is lower than the previous methods due to the careful design.

**Weaknesses:**

1. The restored images seem to be over-smoothed.

**Questions:**

1. Please explain why the computational complexity and memory footprint of the proposed method is lower than the previous method. Is it due to the design of the model or the choice of the straight-line flow?

---

> ### Author Response · Authors · 2024-11-20
> **Response to questions**
>
> Q1. Indeed, our images tend to be slightly oversmoothed, but we do not see it as a weakness, but rather inevitable due to the distortion perception tradeoff. Our method seems to be more on the “distortion” side of the perception/distortion tradeoff (implying high PSNRs and smoother images). As shown in [4], a method cannot consistently yield high perception and low distortion images, therefore one should generally not expect one method to beat all the others in both perceptual and distortion metrics.
>
> Q2. Indeed, our method is the cheapest among the benchmarked flow matching methods. This is because we only need (few) evaluations of the velocity field. D-Flow needs to backpropagate through the ODE solution (very memory intensive) and flow priors needs to compute the trace of the flow’s jacobian. OT-ODE requires to solve a linear inverse problem at each iteration.
>
> [4] Blau, Michaeli, The perception-distortion tradeoff, CVPR 2018

---

> > ### Comment · Reviewer_VJZf · 2024-11-26
> >
> > Thanks a lot for your response. I'm convinced by the method proposed by the authors. In particular, the proposed method is more efficient than the previous method. In addition, it generalizes well to several image restoration tasks. Thus, I decided to keep my rating of this paper.

---

### Official Review · Reviewer_fVXi · 2024-11-04

**Soundness:** 2
**Presentation:** 3
**Contribution:** 3
**Rating:** 6
**Confidence:** 4

**Summary:**

This paper proposes to use flow matching in the plug-and-play framework for image restoration. The key is to use FM model as the denoisier. To avoid the numerical challenges, it integrates the implicit FM prior into a custom denoisier.

**Strengths:**

1, It proposes a design a time-dependent denoiser based on a pre-trained velocity field v learned through Flow Matching

2, This denoiser is integrated into an adapted Forward-Backward Splitting PnP framework that cycles through a gradient step on the data-fidelity term, an interpolation step and a denoising step

3, Being computationally efficient and memory-friendly via the use of ODE

**Weaknesses:**

1, Why the percpetual metrics are missing? From the visual results, it also seems that the results tend to be blurry. What’s the underlying reason? Is it due to the gradient step or the interpolation step, or something else?

2, In addition, one of the advantages of these generative method is its high perceptual quality, but this method seems to have achieved good distortion performance. How about the results of employing the same end-to-end U-Net model as a simple baseline (for example, using the L1 loss)?

3, Can you visualize all the intermidate resutls of all three steps for all time steps? It could better help readers understand the method.

**Questions:**

See weakness

---

> ### Author Response · Authors · 2024-11-20
> **Response to questions**
>
> Q1. We computed the LPIPS metric, see Appendix A.10. We have competitive performance, with no method consistently leading the LPIPS metric. Our method seems to be more on the “distortion” side of the perception/distortion tradeoff. As shown in [4], a method cannot consistently yield high perception and low distortion images, therefore one should generally not expect one method to beat all the others in both perceptual and distortion metrics.
>
> Q2. We think this is an interesting suggestion, however in this paper we want to focus on pretrained flow matching/diffusion/denoising models since they do not rely on paired data and are not task-specific. If we understand your suggestion correctly, a U-Net would need to be retrained for changes in the forward operator or degradation noise model.
>
> Q3. We appreciate the suggestion. A new visualization can be found in Appendix A.9, Figure 13.
>
> [4] Blau, Michaeli, The perception-distortion tradeoff, CVPR 2018

---

> > ### Author Response · Authors · 2024-11-29
> > **Reminder**
> >
> > Dear reviewer fVXi,
> >
> > thank you for your review. We appreciate your feedback. We carefully revised our manuscript according to your concerns. We hope that the revised version **meets your expectations**. If you have any **remaining concerns**, please let us know as soon as possible. If not, we would kindly invite you to consider **raising the score**.
> >
> > Sincerely, the authors

---

### Official Review · Reviewer_n2r9 · 2024-11-04

**Soundness:** 2
**Presentation:** 3
**Contribution:** 2
**Rating:** 5
**Confidence:** 3

**Summary:**

The paper proposes a zero-shot method (PnP-flow) for Inverse problems based on a pre-trained flow-matching (FM) model. The method combines the plug-and-play (PnP) framework with flow matching  by alternating between gradient descent steps on the data-fidelity term, reprojections onto the learned FM path, and denoising. PnP-flow achieves state-of-the-art (SOTA) results compared to existing PnP and flow-based algorithms  across different image inverse problems.

**Strengths:**

1. The method is training-free which makes it computationally practical.
2. The method achieves SOTA results compared to existing flow-based methods.

**Weaknesses:**

1. My major concern is the lack of comparison to recent zero-shot methods based on a pre-trained diffusion model such as DDNM [1] and DPS [2].
2. The proposed method is non-blind (assume the full knowledge of the degradation model) which limits its applicability.


[1] Wang et al. Zero-Shot Image Restoration Using Denoising Diffusion Null-Space Model. ICLR 2023

[2] Chung et al. Diffusion Posterior Sampling for General Noisy Inverse Problems. ICLR 2023

**Questions:**

1. Could you add comparisons with [1] and [2], or explain why those comparison are missing?
2. Could you comment on the potential applicability/extension of your method to the blind case?

---

> ### Author Response · Authors · 2024-11-20
> **Response to questions**
>
> Q1. We added comparisons with DPS and DDRM in Appendix A.13 (Table 10&11). DDNM seems like an very interesting approach for future research: indeed, we could use a similar null space decomposition and only perform our proposed PnP Flow within this null space. We add a discussion on DDNM and related diffusion methods in the appendix, see Appendix A.12.
>
> Q2. Yes, our method is adaptable to the blind case and we thank the reviewer for this interesting suggestion. We demonstrate this on a preliminary experiment on blind deconvolution and blind inpainting where we only assume to have access to the observation and the pre-trained model, without additional data. Then we iterate (over time) between our algorithm steps and update steps on a randomly initialized kernel/mask. We see that the reconstruction is of higher quality than the blurry measurement. See the promising results in Appendix A.14.

---

> > ### Author Response · Authors · 2024-11-29
> > **Reminder**
> >
> > Dear reviewer n2r9,
> >
> > thank you for your review. We appreciate your feedback. We carefully revised our manuscript according to your concerns. We hope that the revised version **meets your expectations**. If you have any **remaining concerns**, please let us know as soon as possible. If not, we would kindly invite you to consider **raising the score**.
> >
> > Sincerely, the authors

---

### Official Review · Reviewer_pWBL · 2024-11-07

**Soundness:** 2
**Presentation:** 1
**Contribution:** 2
**Rating:** 3
**Confidence:** 5

**Summary:**

This paper introduces the PnP-Flow Matching algorithm for addressing imaging inverse problems, including denoising, super-resolution, deblurring, and inpainting. The method combines the Plug-and-Play (PnP) framework with Flow Matching (FM) models by using a time-dependent denoiser to tackle image restoration tasks. Specifically, the algorithm alternates between gradient descent on a data fidelity term, reprojection onto a flow matching path, and denoising. The experiments demonstrate the effectiveness of the proposed method.

**Strengths:**

1. Low memory usage, making it suitable for high-resolution images.
2. Consistently performs well across multiple tasks, showing stable PSNR and SSIM improvements.

**Weaknesses:**

1. The writing quality needs Improvement. Certain explanations lack clarity, particularly in describing the algorithmic process, e.g., the function F.
2. The details of the proposed method are insufficient.
3. The experiment section should be improved. Please refer to the details below.

**Questions:**

1. The formula "y = noisy(Hx)" uses a general definition for the noisy function. It would be helpful if the paper and experiments explored multiple types of noise to assess the method’s robustness.

2. In Tables 1 and 2, the comparison is limited, particularly with only one diffusion-based method, PnP-Diff, which is a workshop paper, not a main conference paper. The authors should include comparisons with more diffusion-based methods, such as DPS, DeqIR, and DDRM, to provide a fuller view of how their method performs relative to the latest diffusion techniques.

3. The authors could enhance the evaluation by including the ImageNet dataset. For the denoising and deblurring tasks. Testing the method across various noise levels and degrees of blur on a large, diverse dataset like ImageNet would offer more insight into how well the algorithm handles different types of degradation.

4. In Figure 3, the visual results do not show a significant improvement over other methods (e.g., in the last row), even though the PSNR scores are higher.

5. For real-world data with unknown degradation, it would be important to understand how well this method generalizes.

6. It would strengthen the paper if the authors included examples of failure cases.

7. In Table 3, not all methods are compared for computational time and memory usage. Including all relevant methods in this comparison would give a clearer picture of how the proposed algorithm stacks up in terms of efficiency across different benchmarks.

---

> ### Author Response · Authors · 2024-11-13
> **Clarification for rebuttal**
>
> Thank you for your review. We will address your comments on the numerical section more thoroughly in our next response. It seems that you found certain parts of the paper unclear, while the other reviewers considered the presentation to be good. Could you kindly provide specific points where the explanation was unclear, and indicate what additional details you believe are necessary? For reference, the function F (which is the data-fidelity term) is defined on line 45, and the steps of the algorithm are detailed in Section 3.2. We look forward to making the necessary revisions to enhance the clarity of the paper.

---

> ### Author Response · Authors · 2024-11-20
> **Response to questions**
>
> Regarding Weakness:
> W1 and W2. Since all other reviewers found the presentation good (rated 3), we would appreciate a clarification on what parts you find need more explanation. Without your feedback, we cannot improve the presentation.
>
> Regarding the questions:
>
> Q1. We implemented a super-resolution task with a Laplacian noise degradation model (which is heavy tailed and therefore challenging). Some of the Flow-Matching based methods were not applicable as they are designed for Gaussian noise (D-Flow, OT-ODE), but we benchmarked against PnP Diffusion, PnP GS, Flow priors. Evaluation of the methods on 100 test images are reported in the Table below and in Appendix A.11 (Table 9) with more details.
> | **Method**        | **PSNR ($\sigma=0.1$)** | **SSIM ($\sigma=0.1$)** | **LPIPS ($\sigma=0.1$)** | **PSNR ($\sigma=0.3$)** | **SSIM ($\sigma=0.3$)** | **LPIPS ($\sigma=0.3$)** |
> |--------------------|-------------------------|--------------------------|--------------------------|-------------------------|--------------------------|--------------------------|
> | **Degraded**       | 8.82                   | 0.086                   | 0.845                   | 8.76                   | 0.040                   | 0.865                   |
> | **PnP-Diff**       | *29.67*                | *0.866*                 | **0.059**               | 24.82                  | *0.730*                 | 0.268                   |
> | **PnP-GS**         | 28.50                  | 0.828                   | 0.079                   | *25.31*                | 0.725                   | *0.185*                 |
> | **Flow-Priors**    | 27.53                  | 0.721                   | 0.102                   | 21.72                  | 0.454                   | 0.376                   |
> | **PnP-Flow (Ours)**| **30.30**              | **0.895**               | *0.063*                 | **26.50**              | **0.809**               | **0.108**               |
>
>
> Q2. We added DDRM and DPS to our benchmark. For the results, see Appendix A.13 (Table 10&11).
>
> Q3. While we agree that this would be a very interesting experiment, we are unfortunately unaware of any publicly available unconditional pretrained flow matching model for ImageNet. Training one from scratch would require resources beyond our current capacity.
>
> Q4. Zooming in the images of Fig. 3 and 4 in the paper or Fig. 7 in the appendix, one can actually see important visual differences, which are discussed in Section 5.3. Note that none of the competing methods performs as consistently across all tasks as our method.
>
> Q5. In the revised version of the paper, we conduct new blind-restoration experiments. We adapt our method PnP-Flow to cases of unknown blur or unknown mask without any additional data, yielding excellent visual results. Details and results have been added in Appendix A.14.
>
> Q6. We did not observe failure cases for our method. As opposed to the benchmarking methods, our method is not prone to artifacts and other kinds of failures, although it tends to produce averaged results (slightly over-smoothed), consistent with its interpretation as a minimum-mean-square-error (MMSE) estimator. Interestingly, a competing method like OT-ODE may generate more realistic textures but often produces highly artifacted and distorted images, as shown in Figure 7 in the appendix.
>
> Q7. Benchmarks were conducted exclusively for Flow-Matching-based methods, as they share the same network architecture. A comparison with diffusion methods would not be relevant here.

---

> > ### Author Response · Authors · 2024-11-29
> > **Reminder**
> >
> > Dear reviewer pWBL,
> >
> > thank you for your review. We appreciate your feedback. We carefully revised our manuscript according to your concerns. We hope that the revised version **meets your expectations**. If you have any **remaining concerns**, please let us know as soon as possible. If not, we would kindly invite you to consider **raising the score**.
> >
> > Sincerely, the authors

---

### Author Response · Authors · 2024-11-20
**General answer to all the reviewers**

We thank all the reviewers for their effort and constructive feedback. All reviewers enjoyed the excellent performance of our method across multiple tasks and its computational efficiency and low memory footprint.

We uploaded a revised version of the paper according to all concerns of the reviewers with modifications appearing in blue. In particular, we made a big effort to add numerical experiments. We address some common/major points in this general answer.

- Based on the suggestion of reviewer pWBL, we added an experiment incorporating Laplace degradation noise. Benchmarking against other methods demonstrates the adaptability of our PnP-flow method. Here, we outperform competing methods across all metrics.
- We added more diffusion baselines, namely DPS [1] and DDRM [2], as recommended by reviewer n2r9 and pWBL.
- As suggested by reviewers pWBL and n2r9, we added numerical experiments for blind inverse problems where the forward is unknown.
- Following the suggestion of reviewer fVXi, we added a perceptual metric, LPIPS [3].

[1] Chung et al, Diffusion Posterior Sampling for general noisy inverse problems, ICLR 2023

[2] Kawar et al, Denoising diffusion restoration models, NeurIPS 2022

[3] Zhang, The unreasonable effectiveness of deep features as a perceptual metric, CVPR 2018

---

### Author Response · Authors · 2024-11-25
**Friendly reminder**

Dear reviewers, since the discussion period is ending soon, we kindly remind you of giving us feedback on the revised version of our paper. We took great care and efforts in revising it and addressed all your concerns.

---

### Author Response · Authors · 2024-12-04
**Discussion Summary**

The review period is now over, and there unfortunately has not been any discussion regarding our paper, although we really tried to engage one.  We believe that we have solid contributions, and adressed the reviewers concerns.
Many thanks to the reviewer, who gave already score 8 and underlined again that she/he believes in our work..

---

> ### Author Response · Authors · 2024-12-04
> **Contributions Summary**
>
> For all the interested readers we list our contributions below:
>
> 1) We proposed a Plug and Play version of Flow Matching, which introduces a time dependent denoiser based on the velocity from flow matching.
>
> 2) We compared extensively against other flow matching (and diffusion) methods, which did not have available code. We make the comparisons public.
>
> 3) We show that our proposed method has theoretical advantages over diffusion (straight paths), and practical advantages (non Gaussian latent).
>
> 4) During the rebuttal, we also added more diffusion baselines, and showed that our method can be applied to blind inverse problems and non-Gaussian noise.

---

### Meta-Review · Area_Chair_62aJ · 2024-12-20

**Metareview:**

Reviewers agree with the practicality of the proposed method and its novelty.
Reviewers pWBL and n2r9 had some concerns about the validation of the model, which the authors addressed and included in their rebuttal. But the reviewer did not further comment on those results.
Reviewer pWBL made further comments about the text clarity and this was again addressed by the author.
Given that the authors responded to reviewer comments and other reviewers were happy with the paper's contribution and evaluation, this paper is recommended for acceptance.

**Additional Comments On Reviewer Discussion:**

Reviewers were not responsive to author feedback.

---

### Decision · Program_Chairs · 2025-01-22

Accept (Poster)